# A neural network model that generates salt concentration memory-dependent chemotaxis in *Caenorhabditis elegans*

**Masakatsu Hironaka[1], Tomonari Sumi[1,2]\*†**

[1]Department of Chemistry, Faculty of Science, Okayama University, Okayama, Japan;
[2]Research Institute for Interdisciplinary Science, Okayama University, Okayama, Japan

**\*For correspondence:**
sumi@muroran-it.ac.jp

**Present address:** †Department of Sciences and Informatics, Muroran Institute of Technology, Muroran, Japan

**Competing interest:** The authors declare that no competing interests exist.

## eLife Assessment

With a computational analysis of a neuroanatomical network model in *C. elegans*, this **valuable** work investigates the synaptic mechanism for memory-dependent klinotaxis, i.e., salt concentration chemotaxis. By incorporating experimental data altering the ASER neuron's basal glutamate release into their model, the authors demonstrate the possibility of a transition between excitatory and inhibitory signaling at the ASER-AIY synapse, depending on environmental and cultivated salt concentrations. These **solid** findings offer a proposal for how synaptic plasticity plays a role in sensorimotor navigation, and will be of interest to worm biologists and theoretical neuroscientists.

**Abstract** A neuroanatomical minimal network model was revisited to elucidate the mechanism of salt concentration memory-dependent chemotaxis observed in *Caenorhabditis elegans. C. elegans* memorizes the salt concentration during cultivation, manifesting a pronounced taste preference for this concentration. The right-side head sensory neuron, designated ASER, exhibits a response to a decrease in salt concentration. The basal level of glutamate transmission from ASER has been demonstrated to transiently increase and decrease when the current environmental salt concentrations are lower and higher, respectively, than that during previous cultivation. Given the sensitivity of excitatory/inhibitory glutamate receptors expressed on the postsynaptic AIY interneurons, it can be anticipated that the ASER–AIY synaptic transmission will undergo a reversal due to alterations in the basal glutamate release. The neural network, derived from the hypothesis, reproduced the salt concentration memory-dependent preference behavior and revealed that the circuit downstream of ASE functions as a module that is responsible for salt klinotaxis.

## Introduction

It is a fundamental biological principle that all animals have evolved the ability to efficiently access the optimal food patches or nesting sites for survival. This ability is achieved by the animals temporarily memorizing the intensity of sensory cues (e.g., temperature, concentration of substances, etc.) that coincide with the favorable environmental conditions they have previously experienced. The neural system should determine a preferred direction based on both the previously memorized sensory information and the change in the stimulus intensity currently received by the sensory neurons, and regulate the motor neurons to efficiently direct the animal to the optimal location in the real time. However, the precise mechanisms by which the intensity of experienced sensory stimuli is memorized or encoded in the neural system and the favorable direction is varied as a function of those sensory cues remain unclear for a considerable number of sensory stimuli.

The nematode *Caenorhabditis elegans* has been demonstrated to exhibit chemotaxis to sodium chloride, a phenomenon termed salt chemotaxis (*Ward, 1973*). In the context of salt chemotaxis, the amphid taste neurons ASE play a crucial role in sensing the concentration of NaCl (*Bargmann and Horvitz, 1991*). These sensory neurons, which consist of two morphologically symmetric neurons on the left (ASEL) and the right (ASER), are essential for the detection of alterations in the concentration of NaCl. These neurons exhibit a functional asymmetry in which the ASER is depolarized by decreases in salt concentration, whereas the ASEL responds to increases (*Suzuki et al., 2008*).

Two distinct behavioral strategies that direct *C. elegans* toward a favorable NaCl concentration during salt chemotaxis have been characterized by an analysis of the migratory behaviors exhibited by the organism (*Iino and Yoshida, 2009*; *Pierce-Shimomura et al., 1999*). The first is klinokinesis, which is also referred to as the pirouette. In this strategy, the frequency of redirecting turns is increased when the current direction of locomotion is determined to be unfavorable in relation to the gradient of salt concentration and decreased when it is determined to be favorable (*Kunitomo et al., 2013*; *Pierce-Shimomura et al., 1999*). The second strategy is referred to as klinotaxis or weathervane. In this behavioral strategy, *C. elegans* exhibits a continuous turning behavior in a direction that is favorable based on the gradient of salt concentration perpendicular to the direction of locomotion (*Iino and Yoshida, 2009*; *Kunitomo et al., 2013*). Furthermore, the salt concentration preferred by *C. elegans* is demonstrated to depend on the difference in salt concentration between the pre-assay culture ($C_{cult}$) and the current environment on the test plate ($C_{test}$). In the presence of food, if the previously cultivated salt concentration is higher than the current environmental concentrations ($C_{cult} > C_{test}$), the preferred salt concentrations are demonstrated to become higher than the current concentrations (*Kunitomo et al., 2013*). Conversely, if the previously cultivated salt concentration is lower than the current concentrations ($C_{cult} < C_{test}$), the preferred salt concentrations are demonstrated to become lower than the current ambient concentrations (*Kunitomo et al., 2013*).

The neural circuit mechanism underlying such salt concentration memory-dependent chemotaxis has been intensively investigated, particularly in the context of klinokinesis (*Hiroki et al., 2022*; *Sato et al., 2021*). The interneuron postsynaptic to the ASER, AIB, which is involved in the behavior of klinokinesis (i.e., the redirecting turn), exhibits a bidirectional response to changes in NaCl concentration in the current environment, dependent on the previously experienced NaCl concentration (*Sato et al., 2021*). The bidirectional neural responses have been demonstrated to be mediated by a change in the basal level of glutamate neurotransmitter released from the ASER (*Sato et al., 2021*). The alteration in the basal glutamate transmission results in a bidirectional response of the postsynaptic AIB due to the distinct sensitivities of the excitatory glutamate receptor GLR-1 and the inhibitory glutamate receptor AVR-14, which are expressed on the AIB (*Hiroki et al., 2022*). These findings demonstrate that the reversal of the redirecting turn behaviors in klinokinesis is attributed to the synaptic plasticity between ASER and AIB neurons, which is altered by cultivated salt concentration. In contrast, the salt concentration memory-dependent mechanism underlying another NaCl chemotaxis, klinotaxis, remains largely unexplored.

*C. elegans* has 302 neurons, all of which have been fully mapped on its connectome (*Cook et al., 2019*; *White et al., 1986*). Although its neural network is relatively simple compared to other multicellular organisms, *C. elegans* exhibits a wide range of behaviors, including locomotion, foraging, feeding, touch withdrawal, and taxis involving smell, taste, and temperature (*Bargmann, 1993*; *de Bono and Maricq, 2005*). *C. elegans* is, therefore, an ideal organism for the detailed investigation of the relationship between neural connectivity and behavior. However, despite these advances in the connectome, the neural connectivity alone is insufficient for a comprehensive understanding of the neural circuit mechanisms underlying behavior. The integration of neural connectivity information with neurophysiological and behavioral observations is essential for adequately simulating the dynamic interactions within neural networks.

A number of neural network models have been proposed to address this issue (*Appleby, 2012*; *Chen et al., 2022*; *Dunn et al., 2004*; *Ferrée et al., 1996*; *Ferrée and Lockery, 1999*; *Izquierdo and Beer, 2013*; *Izquierdo and Lockery, 2010*; *Izquierdo et al., 2015*; *Matsumoto et al., 2024*; *Soh et al., 2018*). A neural network model has been developed based on the neuroanatomical minimal network circuit (see *Figure 1*), which has been demonstrated to reproduce salt klinotaxis (*Izquierdo and Beer, 2013*). The neuroanatomical minimal model is an extension of a previously proposed sensory-motor model that omits interneurons (*Izquierdo and Lockery, 2010*). After the presentation

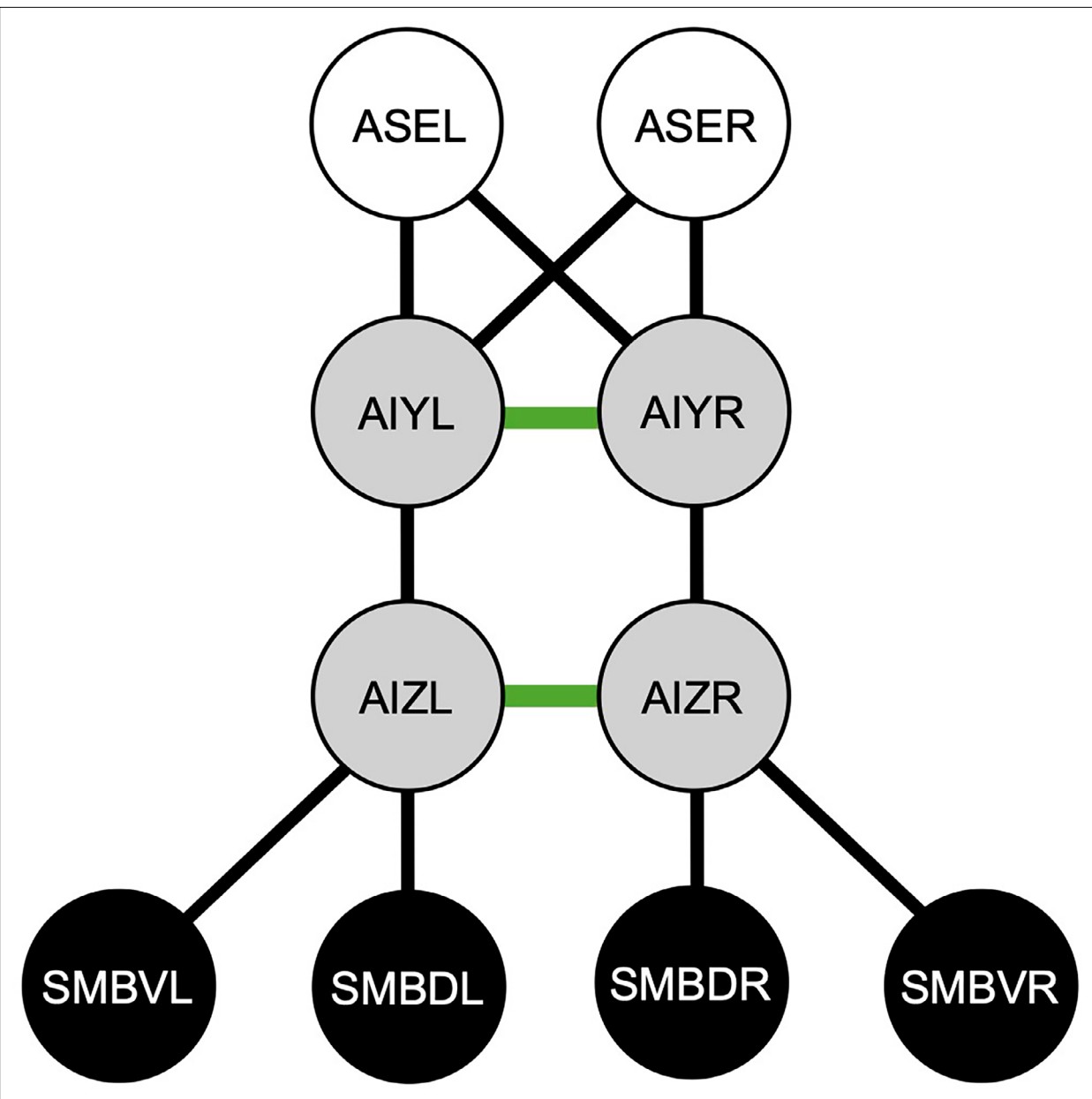

**Figure 1.** A neuroanatomical minimal network circuit for salt klinotaxis in *C. elegans*. The white circles represent chemosensory neurons, the gray circles represent interneurons, and the black circles denote motor neurons. The black and green connections between neurons represent the chemical synaptic connections and electrical gap junctions, respectively. The minimal circuit was derived from the *C. elegans* connectome, with two constraints applied as described in the text (*Izquierdo and Beer, 2013*).

of the model (*Izquierdo and Beer, 2013*), a neurophysiological discovery regarding the synaptic connections between AIY and AIZ interneurons was reported (*Li et al., 2014*). Specifically, it was demonstrated that the AIY plays a crucial role in initiating the curving turn by forming an inhibitory synapse with the AIZ (*Li et al., 2014*). Indeed, it was observed that the inhibitory synaptic transmission is induced when the Cl⁻ ion channel of the acetylcholine (ACh) neurotransmitter receptor, ACC-2, on the postsynaptic AIZ is activated by the ACh neurotransmitter released from the presynaptic AIY (*Li et al., 2014*).

In the present study, the electrophysiological parameters involved in the neuroanatomical minimal network model (*Izquierdo and Beer, 2013*) were re-examined by constraining the AIY–AIZ synaptic transmission to be inhibitory (*Li et al., 2014*). An evolutionary algorithm with the abovementioned constraints was used to extensively search the parameters within the model and optimize the

behavioral performance of the model for salt klinotaxis of *C. elegans* cultivated at a NaCl concentration higher than the current environmental concentrations ($C_{cult} > C_{test}$) in the presence of food. The most optimized model indicated that the ASER is connected to the postsynaptic AIY interneurons via an inhibitory synaptic transmission. The validity of the resulting network was corroborated by the experimental observation that the excitation of the ASER in response to a decrease in NaCl was synchronized with that of the AIZ (*Matsumoto et al., 2024*). Specifically, the synchrony of the excitation of the ASER and AIZ (*Matsumoto et al., 2024*) taken together with the experimentally identified inhibitory synaptic transmission between the AIY and AIZ revealed that the ASER–AIY synaptic connections should be inhibitory, which was consistent with the network obtained from the most optimized model. It was then hypothesized that the ASER–AIY inhibitory synaptic connections are altered to become excitatory due to a decrease in the baseline release of glutamate from the ASER when individuals are cultured under $C_{cult} < C_{test}$. This is due to the substantial difference in the sensitivity of excitatory/inhibitory glutamate receptors expressed on the postsynaptic AIY interneurons. It was postulated that this reversal would result in a reversal of the salt preference behavior observed in klinotaxis. The most optimized model in which solely the ASER–AIY inhibitory synaptic connections were replaced with an excitatory connection, demonstrated a preference for a lower NaCl concentration than the current concentrations, as expected. Finally, the most optimized model was used to investigate the impact of the experimentally suggested reduced activity of the SMB motor neurons in the absence of food on klinotaxis behavior.

## Results

### An evolutionary algorithm discovered a highly chemotaxis-performing network that reproduced the experimentally observed pattern of neuronal activity

Two types of extensive evolutionary searches for the electrophysiological parameters of the neuroanatomical minimal network model were performed by optimizing a chemotaxis index (CI). The first approach was a conventional evolutionary search without additional neurophysiological constraints, as had been employed previously (*Izquierdo and Beer, 2013*). The second evolutionary search was conducted with the constraint that the AIY–AIZ synaptic connections be inhibitory (*Li et al., 2014*). In these genetic algorithms, the networks evolved in chemotaxis assays where conical shapes were employed as salt concentration profiles (*Equation A15*). In contrast, the CI for optimized networks was evaluated using a Gaussian shape of salt concentration profile (*Equation A16*), which mimics

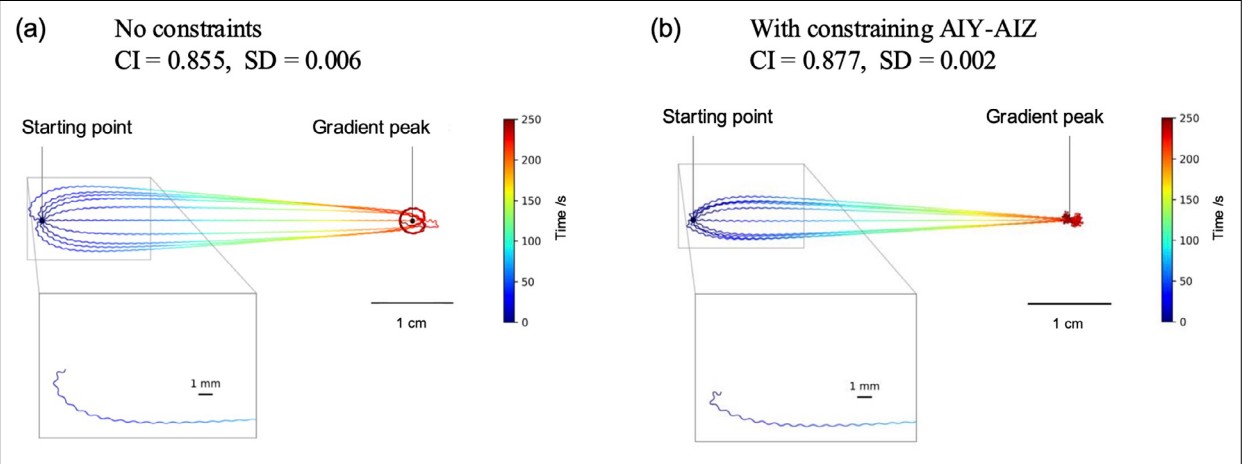

**Figure 2.** The trajectories of the worm's locomotion. The highest-performing network models, with and without the AIY–AIZ connections constrained to be inhibitory, were placed at the initial position, 4.5 cm away from the salt gradient peak, at 10 different angles of worm orientation, and allowed to move freely for 250 s. The salt concentrations were represented by a Gaussian distribution. The color of the trace represents the passage of time. (**a**) The highest-CI network model that evolved without any constraints. The CI is 0.855, with a standard deviation of 0.006. (**b**) The highest-CI network model that evolved with the constraint that the AIY–AIZ connection be inhibitory. The CI is 0.877, with a standard deviation of 0.002. The insets provide an enlarged view of the sinusoidal locomotion and turning processes.

**Video 1.** The video of the worm's locomotion simulated by the most optimized network with the constraints.
https://elifesciences.org/articles/104456/figures#video1

the salt concentration gradients observed in laboratory tests of chemotaxis in *C. elegans* (*Ferrée and Lockery, 1999*; *Ward, 1973*). The CI values for the most optimized networks with and without the neurophysiological constraints were 0.877 and 0.855, respectively, as shown in *Figure 2*. These values were found to be comparable to those reported in the previous study (*Izquierdo and Beer, 2013*). The difference between these CI values is slight, while the model optimized with the constraints exhibits a marginally accelerated attainment of the salt concentration peak, as shown by the trajectories. The slightly higher chemotaxis performance observed in the constrained model is not essentially attributed to the introduction of the AIY–AIZ synaptic constraints but rather depends on the specific individuals selected from the optimized individuals obtained from the evolutionary algorithm. In fact, even when the AIY–AIZ constraints are taken into consideration, the model retains a significant degree of freedom to reproduce salt klinotaxis due to the presence of a substantial parameter space. Consequently, the impact of the AIY–AIZ constraints on the optimization of the CI is expected to be negligible.

The locomotion trajectories of the most optimized network with the constraints are depicted for the cases in which the model worm was oriented at ten different angles at the initial position (*Figure 2b*; see also *Video 1*). For comparison, the trajectories of the most optimized network without the constraints are also depicted in *Figure 2a*. These trajectories show that both the model worms rapidly turned in the direction of the steepest gradient. These model worms exhibited the salt preference behavior in klinotaxis that were consistent with the experimental observations (*Iino and Yoshida, 2009*) and previous simulation studies (*Chen et al., 2022*; *Izquierdo and Beer, 2013*; *Izquierdo and Lockery, 2010*).

Subsequently, in order to determine how the signal arising from alterations in salt concentration was transmitted through the neural circuit to regulate motor systems, we examined the impact of step changes in salt concentration of varying magnitude with different sign on the neurotransmitter release $z_i$ (defined by *Equation A5*) from each neuron. *Figure 3a, b* presents the sign and strength of the weight of the synaptic connection, $w_{ij}$, in the unconstrained and constrained most optimized models employed in this analysis, respectively. *Figure 3c, d* illustrates the results of the analysis of $z_i$ for the most optimized networks where the additional neurophysiological constraints are not considered and are considered, respectively. It is noted that $z_i$ differs from the membrane potential $y_i$ as a consequence of the influence of the bias term $\theta_i$ in *Equation A5*. Moreover, the neuronal activities involved in signal transmission across chemical synapses are primarily detected through $z_i$. In contrast, $y_i$ correlates with the neuronal activity that are involved in gap junction connections, which can be observed experimentally through an increase in intracellular $Ca^{2+}$ concentration. As illustrated in *Figure 3b*, the evolutionary search with the constraints successfully yielded the network with inhibitory AIY–AIZ synaptic connections as the most optimized model. Coincidentally, the evolutionary search without the constraints also yielded the network with inhibitory AIY–AIZ synaptic connections as the most optimized model (*Figure 3a*). However, this is an incidental finding, as the unconstrained evolutionary search also yielded high-CI networks with excitatory AIY–AIZ synaptic connections, which were analogous to the network that had been presented as a representative model presented in the previous study (*Izquierdo and Beer, 2013*). A noteworthy distinction between *Figure 3a and b* is that the ASER–AIY synaptic connections in the constrained model are inhibitory (*Figure 3b*) whereas those in the unconstrained model are excitatory (*Figure 3a*).

Related to these, the signal transmissions between the neurons that are involved in the neural circuit for klinotaxis have been investigated using calcium imaging techniques (*Matsumoto et al., 2024*). As the salt concentration decreased, the activity of the salt-sensing neuron ASER increased, resulting in an observed activation pattern of the interneuron AIZ that was synchronized with the activation of ASER. As illustrated by the red patterns of $z_i$ in *Figure 3d*, the constrained model exhibited a depolarization of AIZ interneurons as the signal output from ASER increased during a decrease in the

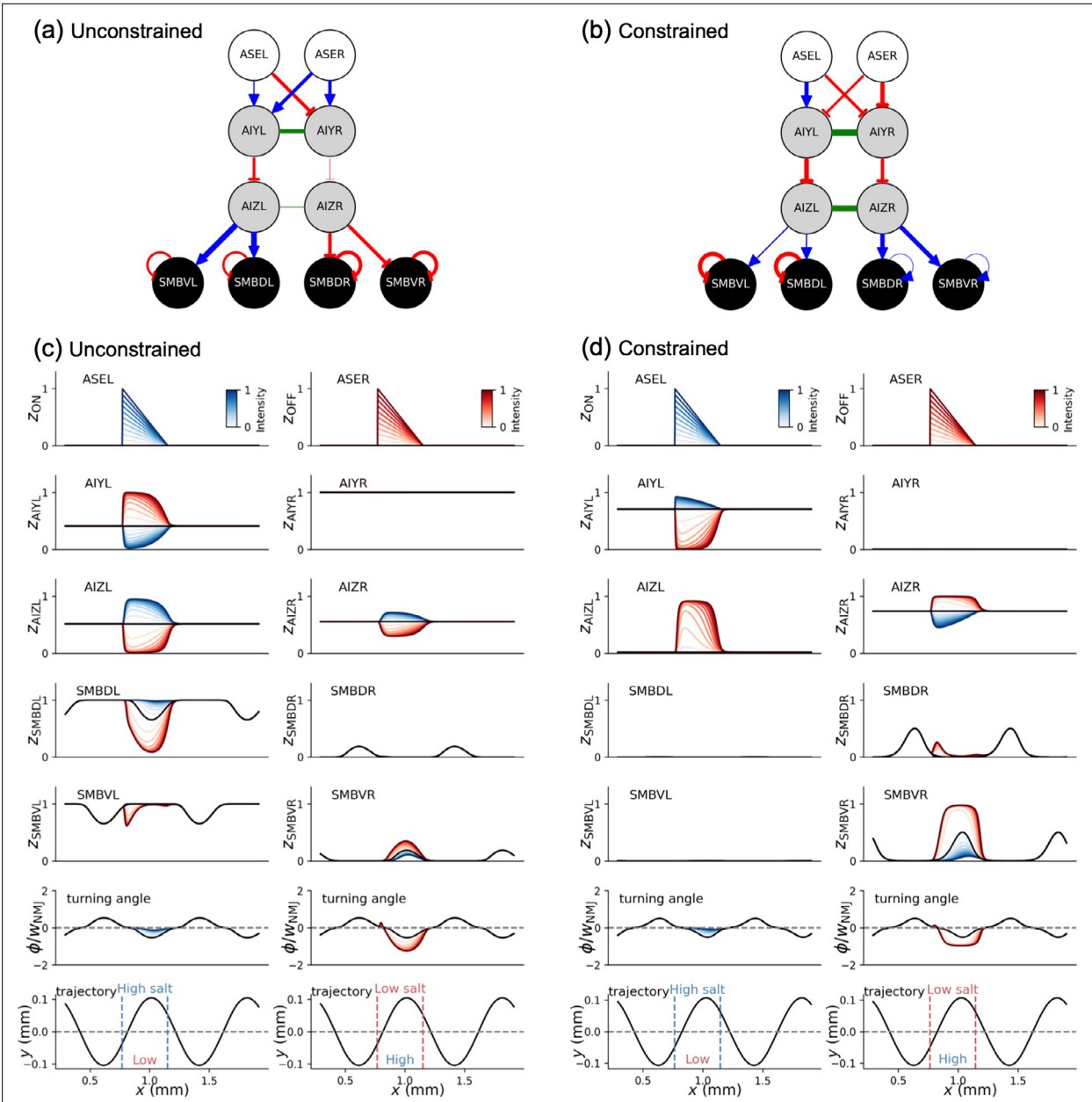

**Figure 3.** The most optimized neural network circuits, with and without constraining the AIY–AIZ connections to be inhibitory, and the resulting network responses to step changes in salt concentration. The most optimized neural network circuits without (**a**) and with (**b**) the constraints. The blue arrow and the red blunt arrow indicate an excitatory and an inhibitory synaptic connection, respectively. The green connections represent electrical gap junctions. The color intensity of these connections indicates the strength of the synaptic connections. (**c**) The neurotransmitter release, $z_i$, from each neuron in the most optimized network without the constraints is illustrated as a response to step changes in the salt concentration. The responses to positive and negative step changes in the salt concentration are represented by the colors blue and red, respectively. (**d**) The illustration is the same as (**c**), but the outcome was obtained from the most optimized network with the constraints. In both (**c**) and (**d**), the black horizontal line represents the level of the bias term, $\theta_i$, in the interneurons and motor neurons. In both (**c**) and (**d**), the turning angle $\varphi$, as defined by *Equation A13a* (see Figure 8b), is illustrated in the second panel from the bottom. In order to identify the direction of the sweep of the head sensory neurons upon introducing step changes in salt concentration, it is necessary to confirm the ideal sinusoidal trajectory of the model worm in the absence of sensory input. This is illustrated in the lowest panel from the bottom in (**c, d**).

The online version of this article includes the following figure supplement(s) for figure 3:

**Figure supplement 1.** Signal transmission through chemical synaptic connections and electrical gap junctions.

**Figure supplement 2.** Blocking electrical gap junctions significantly has a marked effect on neurotransmitter release $z_i$ and the resulting turning angle $\varphi$.

*Figure 3 continued on next page*

*Figure 3 continued*

**Figure supplement 3.** The changes in the turning angles $\varphi$ in response to step changes in the salt concentration of positive and negative at the time half a cycle later than those shown in *Figure 3*.

salt concentration. This indicates that the model reproduces the pattern of neuronal activity observed in the experiments. In contrast, the unconstrained model exhibited an inverse activity pattern of AIZs, namely, a hyperpolarization of AIZs, in response to a decrease in the salt concentration (*Figure 3c*). Nevertheless, the CI performance in the unconstrained model was as high as that in the constrained model. It is therefore essential to consider not only the connectome but also the available neurophysiological information in order to identify potential neural circuits that are consistent with the actual worm among the high-CI networks obtained by the evolutionary search. The inhibitory connections between ASER and AIYs are essential for synchronizing the activation of AIZs with the activity of ASER, provided that the AIY–AIZ connections are inhibitory, as has been identified through experiments. (*Li et al., 2014*; *Matsumoto et al., 2024*). The biochemical mechanism and neurophysiological significance of the ASER–AIY inhibitory connections will be discussed in detail in a subsequent section.

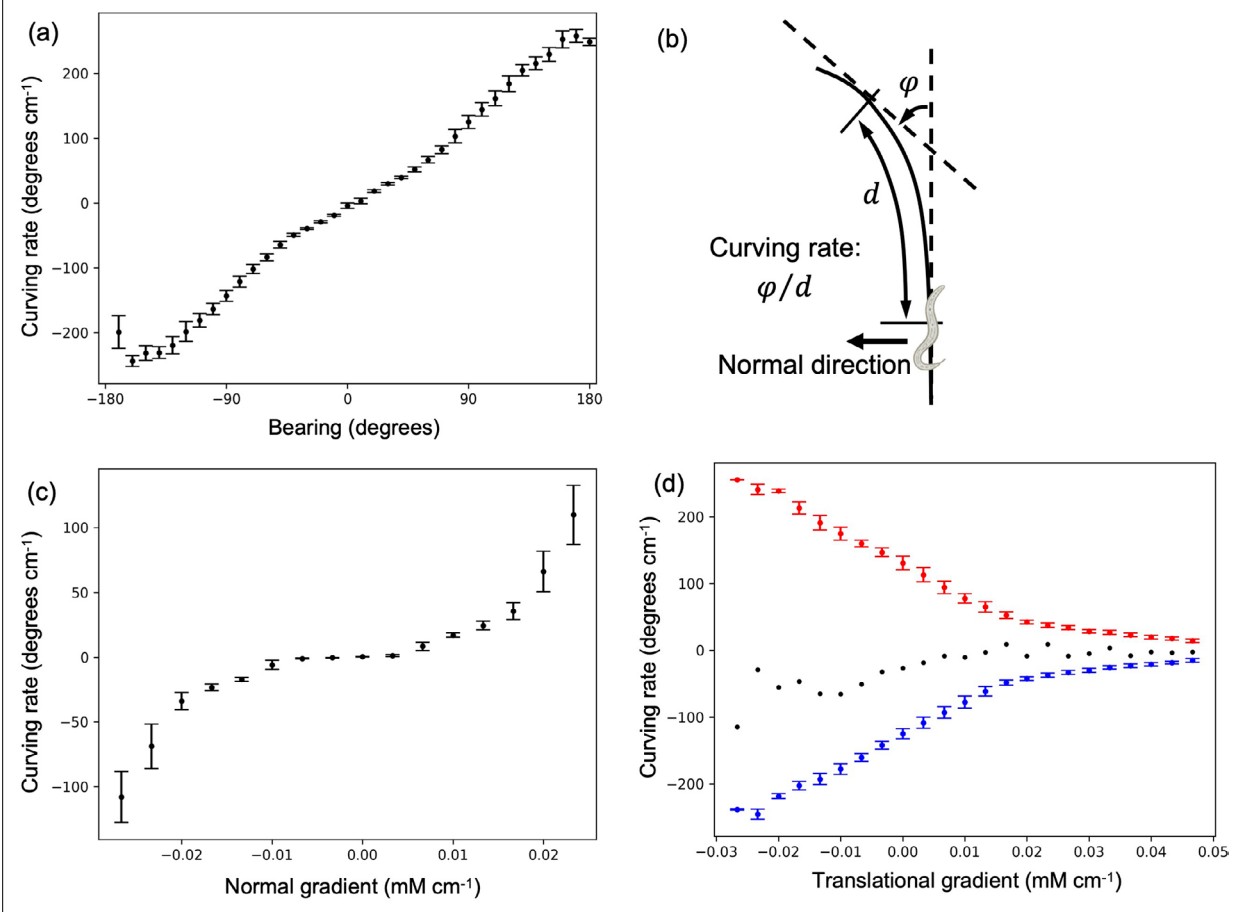

**Figure 4.** The analysis of klinotaxis in the most optimized model with the constraints. (**a**) The curving rate as a function of the bearing. (**b**) The definitions of the curving rate and the normal direction of translational movement, which are utilized in the klinotaxis analysis, are illustrated. (**c**) The curving rate as a function of the normal gradient of salt concentration. (**d**) The positive (red) and negative (blue) components of the curving rate as a function of the translational gradient of salt concentration. The black dots represent the mean value of the two components. In the analysis, the salt concentration profile was modeled with a Gaussian distribution. All the error bars represent the standard deviation.

## The model worm turns to move toward higher salinity based on the concentration gradient normal to its direction of movement

Next, it was investigated how the most optimized model with the constraints reaches the peak of the salt gradient during klinotaxis. To elucidate the mechanism of klinotaxis, the curving rate was examined in relation to the bearing (*Figure 4a*). The curving rate, as defined in *Figure 4b*, was introduced to quantify the behavior of worms during klinotaxis (*Iino and Yoshida, 2009*). The bearing is defined in Figure 9 and described in detail in Bearing in Methods. The positive correlation between the curving rate and the bearing, as illustrated in *Figure 4a*, indicates that the model worm continuously turns in the direction of the steepest increase in salt concentration (see Figure 9). It is, next, necessary to reveal how the model worm determines the direction of its turn in order to increase the salt concentration as it moves. The positive correlation between the curving rate and the normal gradient of the salt concentration, as illustrated in *Figure 4c*, indicates that the model worm is able to sense the concentration gradient normal to the travel direction and thus turn in the direction that increases the salt concentration. These results are consistent with the mechanism that will be proposed later, which indicates that the model worms efficiently sense a slight concentration gradient normal to the direction of travel based on a change in salt concentration with sweeping of the head sensory neurons due to the undulating motions and utilize this information to turn in the correct direction. *Figure 4d* illustrates the positive and negative components of curving rate, as well as the average of these components, as a function of the translational gradient of salt concentration. The translational gradient is defined in Normal and translational salt concentration gradient in Methods. The positive and negative components of the curving rate are, respectively, sampled from the trajectory during leftward turns (as illustrated in *Figure 4b*) and rightward turns, respectively. The average of the positive and negative components of the curving rate was close to zero regardless of the translational gradient, due to the symmetry between the dorsal and ventral regions of the model worm. In contrast, the magnitude of the positive and negative components of the curving rate increased as the translational gradient varied from positive to negative. This indicates that, when the translational gradient is negatively large, the model worm regulates the direction of movement with a large positive or negative curving rate based on the normal gradient. Conversely, when the translational gradient is positive, the worm fine-tunes the direction of movement with a small positive or negative curving rate, which is determined by the normal gradient.

## The minimal neural circuits transmit the signals from the sensory input to the motor systems via both the chemical and electrical connections

Next, focusing on the most optimized network with the constraints (*Li et al., 2014*; *Matsumoto et al., 2024*), the signaling pathways in response to step changes in salt concentration are discussed in detail. Changes in the $z_{\mathrm{ON}}$ signal from the ASEL are transmitted to the AIYL via the excitatory connection (*Figure 3d*). The signals indicating increases in $z_i$ from the AIYL are transmitted to the AIZL via the inhibitory connection between AIYL and AIZL displayed in *Figure 3b*, resulting in a hyperpolarization of the membrane potential $y_i$ in the AIZL (see *Figure 3—figure supplement 1f*). The hyperpolarization signals in the AIZL are transmitted to the AIZR via the gap junction (*Figure 3—figure supplement 1d, f*, *Figure 3d*). This is because the neuron dynamics via gap junctions results from the equilibration of the membrane potential $y_i$ of two neurons connected by gap junctions rather than the $z_i$. The signals indicating decreases in $z_i$ from the AIZR are transmitted to the SMBDR and SMBVR via the excitatory connection (*Figure 3d*). Note that the oscillatory components $z_i^{\mathrm{OSC}}$, as defined by *Equations A10a and A10b*, are introduced into the SMBD and SMBV, respectively. Consequently, the baseline components of $z_i$ from the SMBDR and SMBVR are shifted by half a cycle from each other (see the right side of the fourth and third panels from the bottom in *Figure 3d*). The discrepancies between the $z_i$ values from the SMBD and SMBV neurons (see *Equation A13a*) result in an increase in the turning angle $\varphi$ relative to its ideal oscillatory component (black line), as illustrated on the left side of the second panel from the bottom in *Figure 3d*. The implications of the changes in $\varphi$ will be discussed later.

Changes in the $z_{\mathrm{OFF}}$ signal from the ASER are transmitted to the AIYL via the inhibitory connection (*Figure 3d*). The signals indicating decreases in $z_i$ from the AIYL are transmitted to the AIZL via the inhibitory connection displayed in *Figure 3b*, resulting in a depolarization of the membrane potential $y_i$ in the AIZL (*Figure 3d*). The signals of the depolarization in the AIZL are transferred to the AIZR via the gap junction (see *Figure 3—figure supplement 1d, f*, *Figure 3d*). The signals indicating

increases in $z_i$ from the AIZR are transmitted to the SMBDR and SMBVR via the excitatory connection (*Figure 3d*). The disparities between the $z_i$ values from the SMBD and SMBV neurons (see *Equation A13a*) result in a decrease in the turning angle $\varphi$ relative to its ideal oscillatory component (black line), as illustrated on the right side of the second panel from the bottom in *Figure 3d*. It is remarkable that although the signaling pathway in the most optimized network without the constraints (*Figure 3*, *Figure 3—figure supplement 1*) is largely different from that with the constraints, the regulatory mechanisms of $\varphi$ exhibited by these networks are analogous. This is evident from a comparison of the second panels from the bottom in *Figure 3c, d*. As illustrated by the analysis of these signaling pathways, the transmission of signals via electrical gap junctions plays a crucial role in both the neural circuits. Indeed, it was observed in those neural circuits that blocking the gap junctions led to significant changes in neurotransmitter release $z_i$ and resulted in changes in the turning angle $\varphi$ (see *Figure 3—figure supplement 2*).

## The turning direction is regulated based on both the direction of sweep of the head sensory neurons due to the sinusoidal motion of the worm and the timing of sensory signal input

It was then analyzed how changes in $\varphi$ in response to step changes in salt concentration caused the worm to move in the correct direction. From the sinusoidal trajectory of the model worm (the left side of the first panel from the bottom in *Figure 3d*), it can be observed that the head sensory neurons sweep toward the positive $y$-axis direction during step increases in $z_{\mathrm{ON}}$ occurred. This corresponds to a situation where the salt concentrations on the positive side of the $y$-axis are greater than those on the negative side. The turning angle $\varphi$ is increased from its ideal oscillatory component to a value close to zero, causing the model worm to deviate from the ideal sinusoidal trajectory and gradually turn toward higher salt concentrations. Conversely, in the case of step increases in $z_{\mathrm{OFF}}$, the head sensory neurons sweep toward the positive $y$-axis direction, as illustrated in the right lowest panel of *Figure 3d*. This corresponds to a situation where the salt concentrations on the positive side of the $y$-axis are lower than those on the negative side. The turning angle $\varphi$ is reduced from its ideal oscillatory component to be a more negative value, allowing the model worm to turn faster than the ideal sinusoidal trajectory and thus reach higher salt concentrations with greater efficiency. As illustrated by the second panel from the bottom in *Figure 3c*, the most optimized network without the constraints also uses a similar regulatory mechanism of $\varphi$ to efficiently reach the peak of the salt gradient. To verify the applicability of the elucidated regulatory mechanism of $\varphi$ to other situations, the changes in $\varphi$ in response to step increases in $z_{\mathrm{ON}}$ and $z_{\mathrm{OFF}}$ that were delayed by half of the cycle were examined as a further representative case. This ensured that the regulatory mechanism was operating as intended (see *Figure 3—figure supplement 3*). These observations demonstrate that the change in $\varphi$ relative to its ideal oscillatory component is properly regulated based on the direction of the sensory neuron sweeps at the time of receiving ON and OFF signals via the sensory neurons. The regulatory mechanism of $\varphi$ was found to be essentially identical to that reported in the previous studies (*Chen et al., 2022*; *Izquierdo and Beer, 2013*; *Izquierdo and Lockery, 2010*). It is remarkable that this regulatory mechanism derived via the optimization of the CI has been observed in the context of chemotaxis in *Drosophila* larvae chemotaxis (*Wystrach et al., 2016*) and phototaxis in zebrafish (*Wolf et al., 2017*). The principle of operation, in which the dependence of motor responses to sensory inputs on the phase of motor oscillation, therefore, may serve as a convergent solution for taxis and navigation across species.

## Modifications in the basal glutamate release from the ASER have the potential to elicit bidirectional responses from the postsynaptic AIY

The neural networks generated by the evolutionary algorithm using the CI (*Equation A17*) as the fitness function exhibited klinotaxis, with the model worm turning to move toward higher salt concentrations. This salt preference behavior is consistent with the chemotaxis observed in well-fed individuals cultivated at a higher salt concentration than the current environmental concentrations ($C_{\mathrm{cult}} > C_{\mathrm{test}}$). Conversely, experimental observations have shown that well-fed individuals that are cultivated at a lower salt concentration than the current environment ($C_{\mathrm{cult}} < C_{\mathrm{test}}$) exhibit the opposite salt preference behavior (*Kunitomo et al., 2013*). However, the neural circuit mechanism underlying the reversal of preference behavior in klinotaxis remains unclear.

The reversal of chemotaxis, which depends on the cultivated environmental salt concentration, has been investigated in detail with respect to klinokinesis (*Hiroki et al., 2022*; *Sato et al., 2021*). The interneuron postsynaptic to the ASER, AIB, which is involved in the behavior of klinokinesis, exhibits the bidirectional responses to changes in salt concentration in the current environment, depending on the cultivated salt concentration (*Sato et al., 2021*). The bidirectional responses of the AIB are found to be attributed to a change in the basal level of glutamate neurotransmitter released from the ASER (*Sato et al., 2021*). Specifically, the differential sensitivities of the excitatory glutamate receptor GLR-1 and the inhibitory glutamate receptor AVR-14, which are expressed on the AIB, result in the bidirectional response of the postsynaptic AIB depending on the changes in the basal level of glutamate release from the ASER (*Hiroki et al., 2022*). The GLR-1 is a glutamate-gated cation channel, whereas the AVR-14 is a glutamate-gated chloride anion channel. In addition, the sensitivity of GLR-1 ($EC_{50}$ 5 mM) is several tens of times lower than that of the AVR-14 ($EC_{50}$ 0.2 mM) (*Hiroki et al., 2022*). Consequently, the synaptic transmission between the ASER and AIB is excitatory and inhibitory, respectively, when the basal level of glutamate release from the ASER is high and low. In addition, the biochemical mechanism of the change in the basal level of glutamate neurotransmitter released from the ASER was also elucidated. When the cultivated salt concentration exceeds the current environmental concentrations ($C_{cult} > C_{test}$), the basal glutamate release from the ASER is enhanced by the phosphorylation of UNC-64/Syntaxin 1A at Ser65 through the protein kinase C (PKC-1) signaling pathway (*Hiroki et al., 2022*). Conversely, in the case of $C_{cult} < C_{test}$, the basal glutamate release is decreased as a consequence of the reduced PKC-1 activity (*Hiroki et al., 2022*). The AIB and the neurons further downstream of the AIB show the salt concentration memory-dependent bidirectional responses that correlate with the reversal of the salt preference behavior in klinokinesis (pirouettes). These observations suggest that the reversal of salt preference behavior in klinokinesis is attributed to the synaptic plasticity, which varies from the excitatory to inhibitory synaptic connections between ASER and AIB.

Given those experimental findings, we propose a potential mechanism that may underlie the salt concentration memory-dependent reversal of preferential behavior in klinotaxis. At least two glutamate receptors, the inhibitory glutamate-gated chloride anion channel GLC-3 (*Ohnishi et al., 2011*) and the excitatory metabotropic glutamate receptor MGL-1 (*Kang and Avery, 2009*), have been demonstrated to be expressed in the AIY, the postsynaptic interneuron to the ASER. The $EC_{50}$ for GLC-3 has been reported to be 1.9 mM (*Horoszok et al., 2001*), while the $EC_{50}$ for MGL-1 is not available in the literature. On the other hand, the $EC_{50}$ for the *C. elegans* homolog of MGL-1, MGL-2, has been reported to be 9 μM (*Tharmalingam et al., 2012*). The sensitivity of MGL-2 is ~200 times higher than that of GLC-3, suggesting that the sensitivity of MGL-1 may also be sufficiently higher than that of GLC-3. In the case of $C_{cult} > C_{test}$, if the basal level of glutamate release from the ASER is nearly comparable to the level of $EC_{50}$ for the GLC-3, the activity of MGL-1 is saturated so that the MGL-1 does not contribute to a change in the membrane potential of the AIY upon the depolarization of the ASER (for more details regarding the release of internal $Ca^{2+}$ from the endoplasmic reticulum, see, e.g., the reference on a metabotropic acetylcholine receptor; *Sumi and Harada, 2023*). Thus, an inhibitory synaptic transmission between the ASER and the AIY results from the influx of $Cl^-$ into the cytosol of the AIY via the GLC-3 upon depolarization of the ASER due to a decrease in salt concentration. Conversely, in the case of $C_{cult} < C_{test}$, when the basal level of glutamate release from the ASER is sufficiently lower than the level of $EC_{50}$ for the GLC-3 but still nearly comparable to the level of $EC_{50}$ for the MGL-1, the synaptic transmissions between the ASER and the AIY become excitatory as a result of the $Ca^{2+}$ release from the endoplasmic reticulum, which is mediated by the activation of MGL-1 (*Ashida et al., 2019*; *Shidara et al., 2013*). A similar increase in $Ca^{2+}$ in the AIY was also observed in a previous study on a thermotaxis in *C. elegans* (*Ohnishi et al., 2011*; *Ohta and Kuhara, 2013*), although the precise mechanism remains unclear. On the other hand, it is noteworthy that the biochemical mechanism proposed here does not contradict the ASER–AIY inhibitory connections, as identified in the most optimized network with the constraints.

## Synaptic plasticity between ASER and AIY results in the reversal of salt preference behavior in klinotaxis

In light of the above argument, we investigated the impact of the salt concentration memory-dependent synaptic plasticity between ASER and AIY on the salt preference behavior in klinotaxis. To

this end, we used the most optimized network with the constraints, in which we varied the synaptic connections between ASER and AIY from inhibitory to excitatory. *Figure 5a, b* illustrates the network model with the inhibitory (equivalent to *Figure 3b*) and excitatory connections, respectively. The intermediate states between *Figure 5a and b* are illustrated in *Figure 5—figure supplement 1*. *Figure 5c* shows the curving rates obtained from the most optimized network with the inhibitory (*Figure 5a*) and excitatory (*Figure 5b*) connections as a function of the normal gradient of salt concentration. For comparison, the curving rate obtained from the ninth intermediate connection (see #9 in *Figure 5—figure supplement 1*), which is derived by introducing an increment of 1.5 in the synaptic weight $w_{ji}$ between ASER and AIY at each step, is also presented in *Figure 5c* (also shown as #9 in *Figure 5—figure supplement 2*). Note that the most optimized model with the inhibitory connections (see *Figure 5a*, also shown as #0 in *Figure 5—figure supplement 1*) is considered as an individual that is well fed and cultivated at a higher salt concentration than the current environment. The curving rate was varied from an increasing trend function with increasing the normal concentration gradient to a function with a decreasing trend (*Figure 5c*, see also *Figure 5—figure supplement 2*) when the weight of the ASER–AIY synaptic connection $w_{ji}$ was increased from a negative (inhibitory) to a positive (excitatory) value. These findings are consistent with the experimental observation shown in *Figure 5d*, indicating that the most optimized network with the constraints, of which only the inhibitory connections between ASER and AIY were replaced by excitatory connections (*Figure 5b*), can reproduce the salt preference behavior of a well-fed individual cultivated at a lower salt concentration than the current environment.

A similar analysis to that displayed in *Figure 3d* is presented in *Figure 5e*, with the exception that the inhibitory connections between ASER and AIY have been replaced by the excitatory connections as illustrated in *Figure 5b*. Note that the changes in $z_i$ and $\varphi$ in response to step changes in $z_{\mathrm{ON}}$ (depicted in blue) observed in the neurons further downstream of the ASEL are fully identical to those illustrated in *Figure 3d*, as the synaptic connections between ASEL and AIY remain unaltered. The direction of changes in $z_i$ and $\varphi$ in response to step changes in $z_{\mathrm{OFF}}$, as indicated in red, accords with that of $z_i$ and $\varphi$ in response to step changes in $z_{\mathrm{ON}}$, as shown in blue (*Figure 5e*). This observation indicates that even when either $z_{\mathrm{OFF}}$ or $z_{\mathrm{ON}}$ increases during a head sweep, the $\varphi$ is varied from its ideal oscillatory component to zero. This causes the model worm diverging from the ideal sinusoidal trajectory and gradually turning in the direction of the head sweep that occurred during the ON or OFF signal. More specifically, the model worm gradually turns in the direction of either the lower or higher side of the salt concentration upon receiving the OFF or ON signal, respectively (see both the first and second panels from the bottom in *Figure 5*, *Figure 5—figure supplement 3*). Nevertheless, as evidenced by the curving rate in *Figure 5c*, the model worm clearly shows a preference for lower salt concentrations in klinotaxis. Therefore, the behavior of turning to lower salt concentrations upon increasing $z_{\mathrm{OFF}}$ should play a more decisive role in the reversal of the salt preference behavior than the behavior of turning toward higher salt concentrations upon increasing $z_{\mathrm{ON}}$. Indeed, as illustrated in *Figure 5—figure supplement 4b*, the majority of trajectories yielded by the neural circuit shown in *Figure 5b* demonstrated that the worm model turned to move in the opposite direction of the peak of the salt concentration gradient (see also *Video 2*). However, a subset exhibited a slight curve toward the gradient peak, which then proceeded straight and passed without any discernible response (*Figure 5—figure supplement 4b* and *Video 2*). These behaviors can be interpreted in terms of the changes in the turning angle φ in response to step increases in $z_{\mathrm{ON}}$ and $z_{\mathrm{OFF}}$ (*Figure 5*, *Figure 5—figure supplement 3*). In the case of step increases in $z_{\mathrm{OFF}}$ as illustrated in the second right panel from the bottom in *Figure 5e*, the turning angle φ is increased from its ideal oscillatory component to a value close to zero, causing the model worm to deviate from the ideal sinusoidal trajectory and gradually turn toward lower salt concentrations. On the other hand, in the case of step increases in $z_{\mathrm{ON}}$ as illustrated in the second left panel from the bottom in *Figure 5e*, the turning angle φ is again increased from its ideal oscillatory component to a value close to zero, causing the model worm to deviate from the ideal sinusoidal trajectory and gradually turn toward higher salt concentrations. The behaviors that are consistent with these analyses are observed in the trajectory illustrated in *Figure 5—figure supplement 4b*. For a detailed analysis of signal transmission through chemical synaptic connections and electrical gap junctions related to these behaviors, *Figure 5—figure supplement 5* is presented. In summary, the reversal of salt preference behavior in klinotaxis of a well-fed individual cultivated at a lower salt concentration than the current concentrations can be

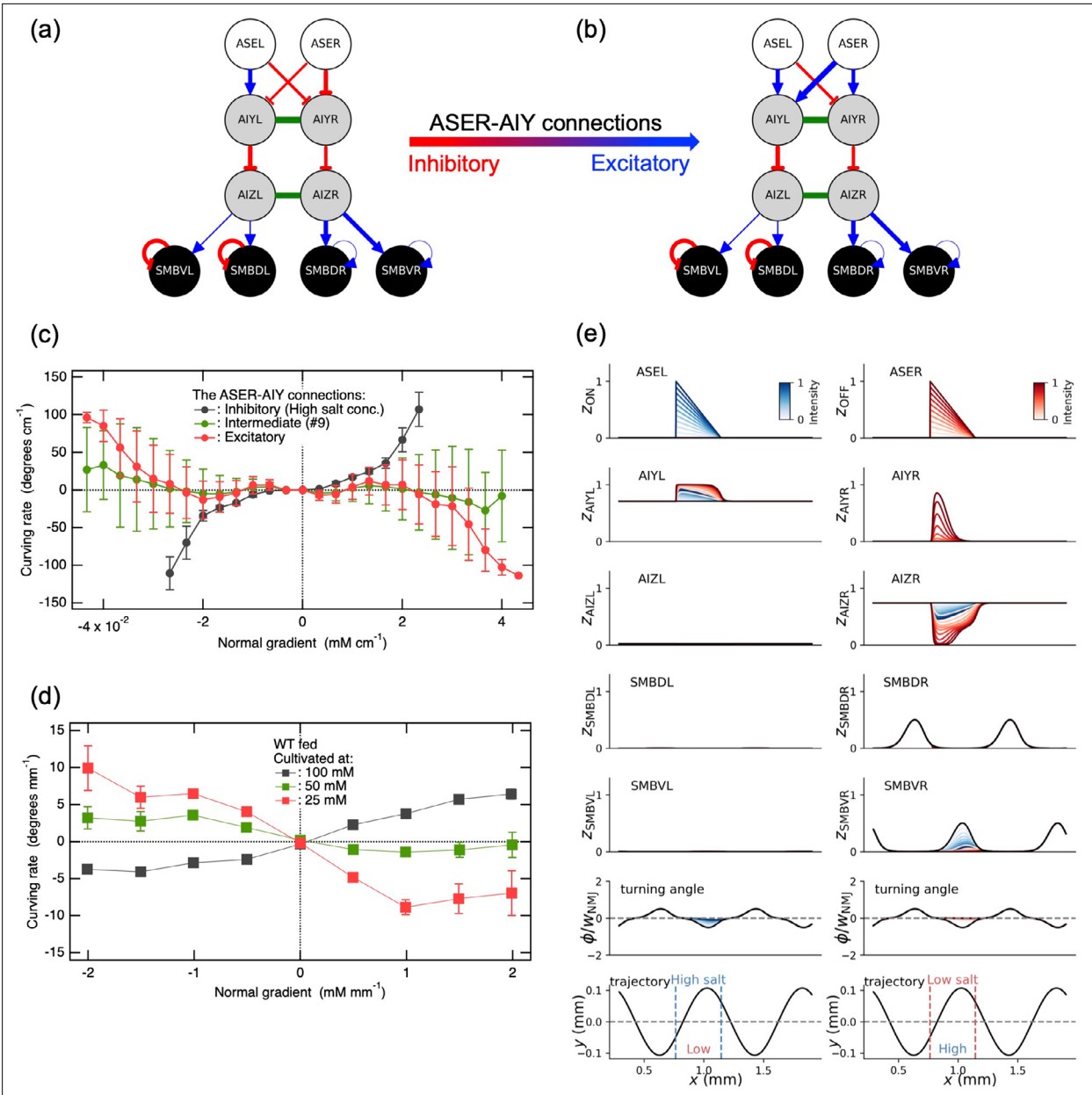

**Figure 5.** The reversal of salt concentration memory-dependent preference behavior in klinotaxis is attributed to the alteration from inhibitory to excitatory connections between ASER and AIY. In the most optimized neural circuit with the constraints, it was postulated that the synaptic connections between ASER and AIY would be altered from (**a**) inhibitory connections to (**b**) excitatory connections when the cultivated salt concentration was replaced from (**a**) a higher to (**b**) a lower concentration than the current environment. The figure shown in (**a**) is identical to *Figure 3b* and #0 of *Figure 5—figure supplement 1*. (**c**) The curving rates obtained from the networks illustrated in (**a**) and (**b**) (corresponding to #0 and #15 in *Figure 5—figure supplements 1 and 2*, respectively), are presented as a function of the normal gradient of salt concentration. In addition, the curving rate obtained from the neural circuit with an intermediate nature in the neural circuit between those shown in (**a**, **b**) (corresponding to #9 in *Figure 5—figure supplements 1 and 2*) is also presented. (**d**) The curving rates as a function of the normal gradient of salt concentration, which were experimentally determined when the cultivated salt concentration was higher (black) and lower (red) than the current environment (*Kunitomo et al., 2013*). The case in which the cultivated salt concentration was close to the current environmental concentration (50 mM) is also shown in green. (**e**) The analysis presented here is identical to *Figure 3d*, except that the ASER-AIY inhibitory connections in the most optimized model with the constraints have been replaced with excitatory connections as illustrated in **b**.

The online version of this article includes the following figure supplement(s) for figure 5:

**Figure supplement 1.** The weight of the ASER–AIY synaptic connection, $w_{ji}$, in the most optimized network with the constraints is increased from a negative value (inhibitory connection) to a positive value (excitatory connection) by introducing an increment of 1.5 to $w_{ji}$ at each step.

*Figure 5 continued on next page*

*Figure 5 continued*

**Figure supplement 2.** As the weight of the ASER–AIY synaptic connection $w_{ji}$ is increased from negative (inhibitory connection) to positive (excitatory connection) in the most optimized network with the constraints by introducing an increment of 1.5 to $w_{ji}$ at each step, the curving rate is varied from an increasing function with an increasing normal gradient of salt concentration to a function that shows a decreasing trend.

**Figure supplement 3.** The changes in the turning angles $\varphi$ in response to step changes in the salt concentration of positive and negative at the timing half a cycle later than those shown in *Figure 5e*.

**Figure supplement 4.** The trajectories of the worm's locomotion simulated by the network shown in *Figure 5a, b*.

**Figure supplement 5.** Signal transmission through chemical synaptic connections and electrical gap junctions.

primarily attributed to the change in synaptic connections between ASER and AIY from inhibitory to excitatory connections.

## Inhibition of SMB activity results in a dispersal behavior that favors starving individuals in their search food

For *C. elegans*, finding food and dwelling at the food source are essential survival strategies. In the absence of food, the release of dopamine from PDE neurons is suppressed, the AVK interneurons, which make inhibitory connections with the PDE, in turn results in the release of FLP-1 neuropeptides (*Oranth et al., 2018*). As a result, the FLP-1 neuropeptides inhibit SMB motor neurons (*Oranth et al., 2018*), which, in conjunction with the reversal of starvation-induced klinokinesis (*Kunitomo et al., 2013*), may result in the dispersal behavior observed in starved individuals. The dispersal behavior of starved individuals, which is necessary for searching distant food, has been reported to manifest as alterations in salt preference behavior in klinotaxis as well as klinokinesis (*Kunitomo et al., 2013*). In particular, the salt concentration memory-dependent preference behavior observed in well-fed individuals (see *Figure 5d*) is markedly suppressed in starved individuals, as illustrated in *Figure 6a* (*Kunitomo et al., 2013*). This implies that in order to efficiently reach distant food sources, the worm suppresses turning and tends to move in a straight line. Given the above neurophysiological findings, we investigated the impact of inhibiting SMB activity on the salt preference behavior in klinotaxis. To this end, we employed the neural circuits in which the ASER–AIY transmission was inhibitory, excitatory, and intermediate between the two, that is, the three circuits that yielded the results shown in *Figure 5c*. *Figure 6b* shows that the inhibition of the SMB activity by reducing the strength of synaptic connections between AIZ and SMB resulted in the suppression of salt preference behavior, which is consistent with the experimental observations presented in *Figure 6a*. Moreover, the inhibition of SMB function through the reduction of bias terms in the SMB motor neurons resulted in the suppression of the salt preference behavior (see *Figure 6—figure supplement 1b*), which is a comparable result to that shown in *Figure 6b* and is also consistent with the experimental data presented in *Figure 6a* or *Figure 6—figure supplement 1a*; *Kunitomo et al., 2013*. A comparison of *Figure 6c, d* (or *Figure 6—figure supplement 1c, d*) with *Figure 5—figure supplement 4a, b*, respectively, showed that the neural circuit models, in which the SMB functions were suppressed, yielded weak spreading behaviors (see also *Video 3*).

## Discussion

In the present study, the neuroanatomical minimal network model was revisited to optimize the electrophysiological parameters using the evolutionary algorithm, where it is important to note that the AIY–AIZ connections were constrained to be inhibitory, as elucidated by experiments (*Li et al., 2014*). The most optimized network, constrained by the abovementioned conditions, successfully reproduced the synchronization of the ASER and AIZ activation patterns observed experimentally in well-fed individuals cultivated at a higher salt concentration than the current environment (*Matsumoto et al., 2024*). The

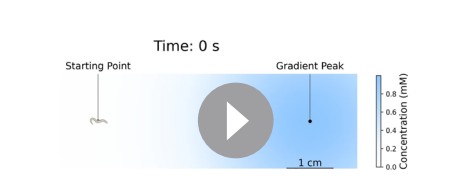

**Video 2.** The video of the worm's locomotion simulated by the network shown in *Figure 5b*.
https://elifesciences.org/articles/104456/figures#video2

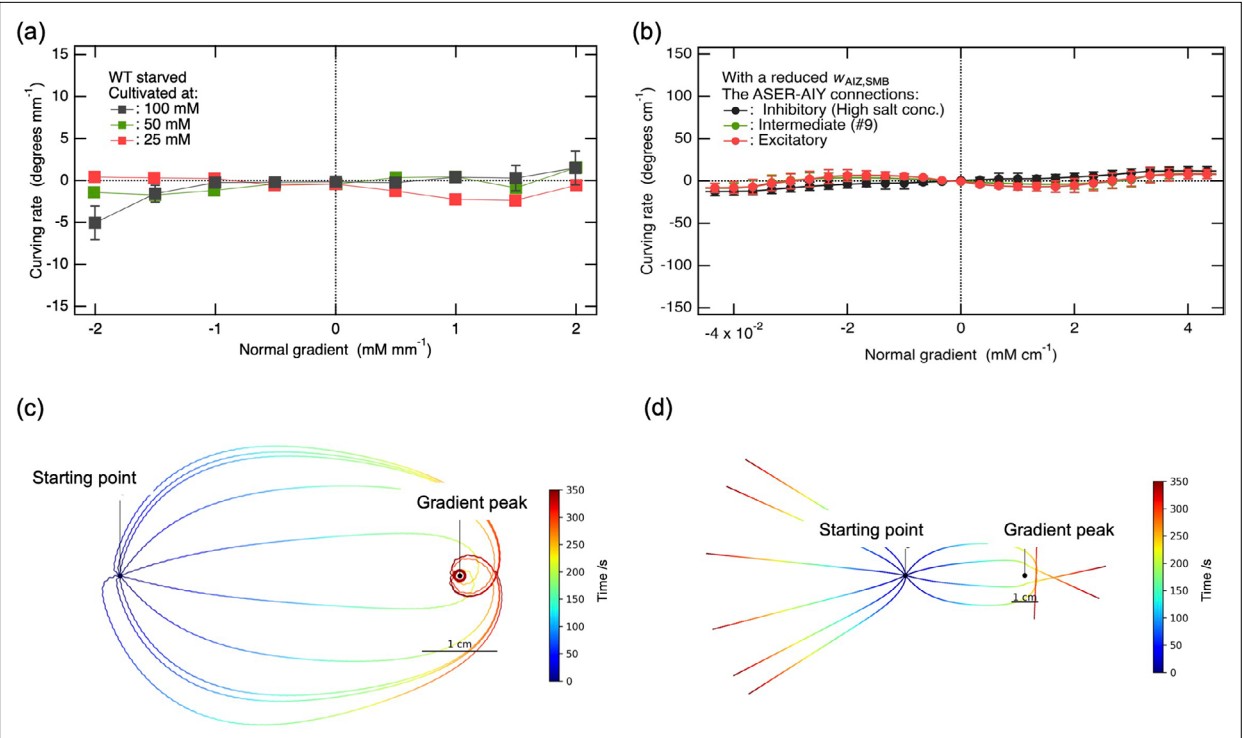

**Figure 6.** Inhibition of SMB activity suppresses the salt concentration memory-dependent preference behavior in klinotaxis, as observed experimentally. (**a**) The curving rates as a function of the normal gradient of salt concentration that were experimentally observed in starved individuals cultivated at a salt concentration higher, comparable, and lower than the current environment (*Kunitomo et al., 2013*). (**b**) The curving rates as a function of the normal gradient of salt concentration that were obtained from the neural circuits that yielded the results in *Figure 5c*, except that here the synaptic connections between AIZ and SMB and the self-connections of SMB were multiplied by 0.9 to inhibit the SMB activity. (**c**) The trajectories of the model worm obtained by inhibiting the SMB activity in the most optimized network, where the ASER–AIY connections remained inhibitory, as shown in *Figure 5a* (or #0 of *Figure 5—figure supplement 1*). (**d**) The trajectories of the model worm obtained by inhibiting the SMB activity in the most optimized network, where the ASER–AIY connections were altered to be excitatory, as shown in *Figure 5b* (or #15 of *Figure 5—figure supplement 1*).

The online version of this article includes the following figure supplement(s) for figure 6:

**Figure supplement 1.** The inhibition of SMB activity by reducing the bias term $\theta_{SMB}$ has the effect of suppressing the salt memory-dependent preference behavior observed in klinotaxis, similar to that illustrated in *Figure 6b*.

inhibitory synaptic connections between ASER and AIY were found to be essential for the synchronization of ASER and AIZ activation patterns, as demonstrated in the most optimized network. The

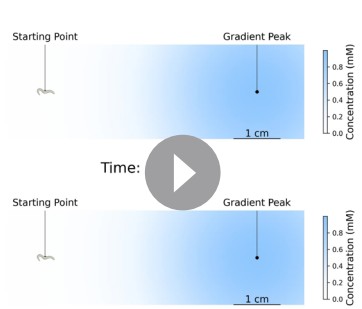

**Video 3.** The videos of the worm's locomotion displayed in *Figure 6c and d* are shown on the upper and lower, respectively.

https://elifesciences.org/articles/104456/figures#video3

mechanism of the ASER–AIY inhibitory connections can be interpreted based on the expression of two glutamate receptors on the AIY, the postsynaptic interneuron of the ASER (*Ohnishi et al., 2011*): the inhibitory glutamate-gated chloride anion channel GLC-3 and the excitatory metabotropic glutamate receptor MGL-1 (*Kang and Avery, 2009*). Based on the $EC_{50}$ values for GLC-3 (*Horoszok et al., 2001*) and MGL-2 (*Tharmalingam et al., 2012*), the *C. elegans* homolog of MGL-1, the sensitivity of MGL-1 to glutamate is expected to be sufficiently higher than that of GLC-3. Given these facts, it can be proposed that if the basal level of glutamate release from the ASER is close to the $EC_{50}$ for GLC-3, the influx of $Cl^-$ into the cytosol of the AIY via the activation of the GLC-3 upon glutamate release from the

ASER due to a decrease in the salt concentration will result in the inhibitory synaptic transmission between the ASER and the AIY. It is remarkable that the inhibitory connections derived from these biochemical arguments are consistent with the abovementioned neurophysiological conclusion, which has been demonstrated by the most optimized network with the constraints. The argument presented here is also supported by evidence that a similar inhibitory synaptic transmission to the AIY due to the influx of $Cl^-$ via the GLC-3, which is activated by glutamate signals from the AFD sensory neurons, has been demonstrated in the context of thermotaxis in *C. elegans* (*Ohnishi et al., 2011*).

In the previous studies on the salt concentration memory-dependent bidirectional regulation of klinokinesis (i.e., pirouettes), it was demonstrated that the basal level of glutamate release from the ASER either increased or decreased when well-fed individuals were cultivated at either a higher or lower salt concentration than the current environment, respectively (*Hiroki et al., 2022*; *Sato et al., 2021*). In light of the experimental findings on klinokinesis and the differential sensitivities of the inhibitory GLC-3 and excitatory MGL-1 that are expressed on the postsynaptic AIY interneurons of the ASER involved in klinotaxis, we proposed a hypothesized mechanism of the salt concentration memory-dependent bidirectional regulation of the preferential behaviors observed in klinotaxis. Specifically, when the well-fed individuals are cultivated at a lower salt concentration than the current environment, and the basal glutamate release from the ASER becomes sufficiently lower than the EC50 for GLC-3 but still nearly comparable to the EC50 for MGL-1, the ASER–AIY inhibitory connections are thought to be altered to become excitatory. This is due to the $Ca^{2+}$ release from the endoplasmic reticulum into the cytosol, which is mediated by the activation of MGL-1. The most optimized network, in which the ASER–AIY inhibitory connections were replaced with excitatory ones, successfully reproduced the reversal of salt preference behavior in klinotaxis that had been previously observed (*Kunitomo et al., 2013*; see *Figure 5c, d*). These findings indicate that the salt concentration memory is encoded as a basal level of glutamate release from the ASER. This memory is then decoded by the difference in the sensitivity of the glutamate-gated inhibitory channel GLC-3 (*Ohnishi et al., 2011*) and the excitatory metabotropic glutamate receptor MGL-1 (*Kang and Avery, 2009*), which are expressed on the AIY. The sensory input from the ASER regarding a reduction in salt concentration is transmitted to the AIY as either an inhibitory or an excitatory signal, depending on the decoded memory. The signaling from the ASER to the AIY is therefore regulated by synaptic plasticity arising from alterations in the basal glutamate release from the ASER. This regulatory process enables the reproduction of salt concentration memory-dependent reversal of preference behavior in klinotaxis, despite the remaining neurons further downstream of the ASER not undergoing alterations and simply functioning as a modular circuit to transmit the received signals to the motor systems. Consequently, the sensorimotor circuit allows a simple and efficient bidirectional regulation of salt preference behavior in klinotaxis.

For individuals experiencing starvation, the expansion of dispersal behaviors is a crucial survival strategy to acquire and dwell distant food sources. Indeed, such behavioral changes have also been observed in the context of salt preference behaviors in klinotaxis (*Kunitomo et al., 2013*). It is therefore necessary to ascertain how the worm's neural network regulates the behavioral strategies in the presence or absence of food. In the absence of food, the PDE neurons do not release dopamine, which causes AVK interneurons that form inhibitory connections with the PDE to release FLP-1 neuropeptides that inhibit SMB motor neurons. This has been postulated to contribute to the dispersal behavior observed in starved individuals (*Oranth et al., 2018*). Therefore, the impact of inhibiting SMB activity on salt preference behavior in klinotaxis was investigated. The most optimized network, in which the strength of the synaptic connections between AIZ and SMB was reduced to suppress SMB activity, while the remainder remained unchanged, showed the suppression of salt preference behavior in klinotaxis (*Figure 6b*), as observed in the experiments (*Kunitomo et al., 2013*; *Figure 6a*). These findings indicate that the worms are capable of efficiently switching between dwelling and dispersing behaviors by activating or inhibiting only the most downstream motor neurons in the sensorimotor circuit of klinotaxis depending on the presence or absence of food.

The principle of operation, in which the dependence of motor responses to sensory inputs on the phase of motor oscillation, appears to be a convergent solution for taxis and navigation across species. In fact, analogous mechanisms have been postulated in the context of chemotaxis in *Drosophila* larvae chemotaxis (*Wystrach et al., 2016*) and phototaxis in zebrafish (*Wolf et al., 2017*). Consequently, the synaptic reversal mechanism highlighted in this study offers the framework for understanding how the

behaviors that are adaptive to the environment are embedded within sensorimotor systems and how experience shapes these neural circuits across species.

## Limitations of the study

The modeling of ASE sensory neurons does not include either the all-or-none depolarization characteristic of the ASEL or the hyperpolarization characteristic of the ASER in response to step increases in salt concentration (*Suzuki et al., 2008*). As a result, a comprehensive understanding of the specific roles played by each the ASEL and ASER could not be achieved. In the present study, the oscillator components of the SMB are not intrinsically generated by an oscillator circuit but are instead externally imposed via $z_i^{\text{OSC}}$. Furthermore, the complex and context-dependent responses of ASER (*Luo et al., 2014*) were not taken into consideration. It should be acknowledged as a limitation of this study that these omitted factors may interact with circuit dynamics in ways that are not captured by the current simplified implementation. The available data on synaptic transmission between the AIZ and the SMB, as well as the available information on the activation pattern of the SMB in response to changes in salt concentration, were insufficient to perform the evolutionary search of parameters involving the SMB with the application of appropriate constraints. Consequently, a comprehensive investigation of the role of the SMB in the regulation of the turning direction was hindered by these limitations.

## Methods

In the present study, a neuroanatomical minimal network model developed by *Izquierdo and Beer, 2013* was employed as the basis for the modeling. The neuroanatomical minimal circuit was derived from the connectome by applying two constraints (*Izquierdo and Beer, 2013*). The first constraint is to minimize the path length between ASE and SMB cells, resulting in a minimum number of three. Since two neurons can be connected by one or more chemical synapses or electrical connections, the total number of neuronal connections was defined as the number of contacts. The second step is to eliminate the interneurons that are connected by neuronal connections other than those with the maximum number of contacts (*Oshio, 2003*). It is noteworthy that the two interneurons, AIY (*Kocabas et al., 2012*) and AIZ (*Iino and Yoshida, 2009*), which have been experimentally shown to be involved in klinotaxis, are included in the resulting minimal network. The primary difference between our model and the neuroanatomical minimal network model is that our modeling constrains the synaptic transmissions between AIY and AIZ to be inhibitory, as experimentally demonstrated (*Li et al., 2014*). A review of the model and computational details is provided in the following section.

### Modeling of sensory neuron signaling

Synaptic transmissions from the chemosensory neurons to interneurons were modeled for the ON (ASEL) and OFF (ASER) sensory neurons, respectively, as an instantaneous function of a coarse-grained time derivative of the recent history of NaCl concentration, $z$ (*Izquierdo and Beer, 2013*):

$$z \equiv \frac{100}{N} \int_{t-N}^{t} dt\, C\left(t\right) - \frac{100}{M} \int_{t-(N+M)}^{t-N} dt\, C\left(t\right), \tag{A1}$$

$$z_{\text{ON}} = \begin{cases} z & (z > 0) \\ 0 & (\text{otherwise}) \end{cases}, \tag{A2}$$

$$z_{\text{OFF}} = \begin{cases} -z & (z < 0) \\ 0 & (\text{otherwise}) \end{cases}, \tag{A3}$$

where $c\left(t\right)$ is the salt concentration at time $t$, and $N$ and $M$ are the durations of the first and second half intervals, respectively, over which the concentration is averaged. The factor of 100 in *Equation A1* was introduced to convert z to values that were more readily amenable to analysis. The ASEL is the ON cell, where the strength of neurotransmitter release $z_{\text{ON}}$ increases in response to an increase in $c\left(t\right)$. Conversely, the ASER is the OFF cell, with $z_{\text{OFF}}$ increasing in response to a decrease in $c\left(t\right)$ (*Suzuki et al., 2008*; see *Figure 7*).

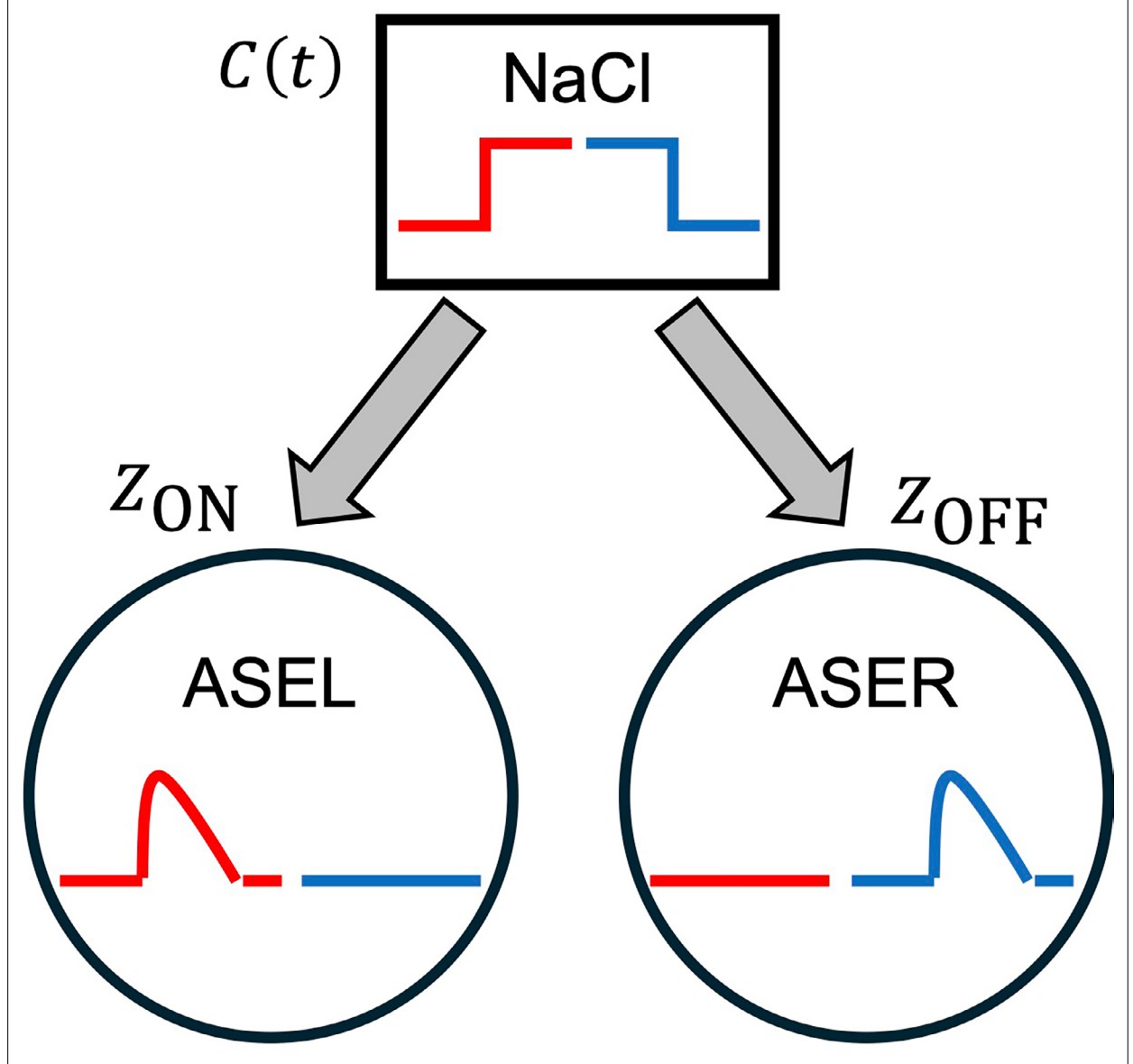

**Figure 7.** Modeling of the synaptic transmission from the ASEL and ASER sensory neurons in response to changes in NaCl concentration.

### Modeling of interneurons

The modeling of interneurons used current-based synaptic transmissions as follows *Izquierdo and Beer, 2013*:

$$\tau_i \frac{dy_i}{dt} = -y_i + \sum_j w_{ji} z_j + \sum_k g_{ki} \left(y_k - y_i\right) + I_i^{\text{IN}}, \quad \left(g_{ki} > 0\right), \tag{A4}$$

where $y_i$ represents the membrane potential relative to the resting potential of zero, $\tau_i$ is the time constant, the first sum represents the input current from chemical synapses, and the second sum represents the input current from electrical synapses. In the first sum, the weight $w_{ji}$ represents the strength of the synaptic connection, and $z_j$ is considered as the strength of neurotransmitter release, as represented by

$$z_i = \sigma \left(y_i + \theta_i\right), \tag{A5}$$

with the following sigmoid function:

$$\sigma\left(x\right) = 1/\left(1 + e^{-x}\right). \tag{A6}$$

$\theta_i$ in *Equation A5* represents the bias term that shifts the range of sensitivity associated with neurotransmitter release. In the second sum of *Equation A3*, $g_{ki}$ represents the conductance of the electrical gap junction, with $g_{ki} > 0$. The fourth term $I_i^{IN}$ in *Equation A3* represents the input current from the chemosensory neurons. This is given by

$$I_i^{IN} = \begin{cases} w_i^{ON} z_{ON} + w_i^{OFF} z_{OFF} & \left(i = \text{AIYL or AIYR}\right) \\ 0 & \left(\text{otherwise}\right) \end{cases}, \tag{A7}$$

where $w_i^{ON}$ and $w_i^{OFF}$ are the strengths of the synaptic connections between interneuron $i$ and the ASEL and ASER sensory neurons, respectively.

## Modeling of neck motor neurons

Neck motor neurons were modeled in a manner analogous to that employed for interneurons, except for the inclusion of self-connections (*Izquierdo and Beer, 2013*):

$$\tau_i \frac{dy_i}{dt} = -y_i + \sum_j w_{ji} z_j + I_i^{MN}, \tag{A8}$$

where the second term represents the input current from chemical synapses, as described in *Equation A5*. The third term represents a constitutive input current from an oscillatory component,

$$I_i^{MN} = w_{OSC} z_i^{OSC}, \tag{A9}$$

where $w_{OSC}$ represents the strength of the connection with the oscillatory component and $z_i^{OSC}$ represents the strength of neurotransmitter release from the oscillatory component. Since the dorsal motor and ventral motor neurons receive a constitutive out-of-phase oscillatory input from the motor systems, respectively, it was postulated that modeling $z_i^{OSC}$ would be achieved by using a sinusoidal function for the dorsal and ventral motor neurons, respectively:

$$z_{SMBDL}^{OSC} = z_{SMBDR}^{OSC} = \sin\left(2\pi t/T\right), \tag{A10a}$$

$$z_{SMBVL}^{OSC} = z_{SMBVR}^{OSC} = \sin\left(2\pi t/T + \pi\right), \tag{A10b}$$

where $T$ represents the duration of a one cycle of sinusoidal motion on agar, which was estimated to be 4.2 s (*Ferrée and Lockery, 1999*).

To reduce the total number of parameters included in the model, in accordance with the neuro-anatomical minimal network model (*Izquierdo and Beer, 2013*), the following constraints were introduced with respect to the second term on the right-hand side in *Equation A8* for the dorsal and ventral motor neurons on the left and right sides, respectively:

$$w_{AIZL,SMBDL} = w_{AIZL,SMBVL}, \ \theta_{SMBDL} = \theta_{SMBVL}, \tag{A11a}$$

$$w_{AIZR,SMBDR} = w_{AIZR,SMBVR}, \ \theta_{SMBDR} = \theta_{SMBVR}, \tag{A11b}$$

These parameters are constrained to be symmetric between the dorsal and ventral motor neurons on the left and right sides, respectively. As discussed in the previous studies (*Izquierdo and Beer, 2013*; *Izquierdo and Lockery, 2010*), a neuron is considered unstable if a self-connection is presented and the weight $w_i^{self}$ is less than 4. This allows the motor neuron with such a self-connection $\left(w_i^{self} < 4\right)$ to respond smoothly to different changes in salt concentration. Similarly, the above symmetry constraints were also applied to the weight $w_i^{self}$ for the self-connections:

$$w_{SMBDL}^{self} = w_{SMBVL}^{self}, \tag{A12a}$$

$$w_{SMBDR}^{self} = w_{SMBVR}^{self}, \tag{A12b}$$

## Worm model

The worm model was constructed using the neuroanatomical minimal network model as the basis for modeling worm locomotion (*Izquierdo and Beer, 2013*). The locomotion of the worm was modeled as the movement of a single point ($r_x$, $r_y$) at a constant velocity $v$ (*Figure 8a*). The angle of the direction of motion $\mu$ relative to the x-axis was measured with counterclockwise being positive (*Figure 8b*). The model involves two assumptions regarding the locomotion mechanism. The first is that the length of the neck muscles is proportional to the strength of neurotransmitter release from the neck motor neurons (*Figure 8a*). The second is that the turning angle $\varphi$ (*Figure 8b*) is proportional to the difference in the length of the neck muscles. According to these assumptions, the equation of motion governing the angle of movement direction $\mu$ is given by

$$\varphi = \frac{d\mu}{dt} = w_{NMJ} \left( \sum_{i_D} z_{i_D} - \sum_{i_V} z_{i_V} \right), \tag{A13a}$$

$$z_{i_D} = \sigma \left( y_{i_D} + \theta_{i_D} \right), \tag{A13b}$$

$$z_{i_V} = \sigma \left( y_{i_V} + \theta_{i_V} \right). \tag{A13c}$$

The summations in *Equation A13a* are performed over the indices $i_D \in \{\mathrm{SMBDL}, \mathrm{SMBDR}\}$ and $i_V \in \{\mathrm{SMBVL}, \mathrm{SMBVR}\}$, respectively. The $w_{NMJ}$ represents the strength of the connection from motor neurons to muscles. In *Equation 13b*, *Equation 13c*, $y_{i_D}$ and $y_{i_V}$ represent the membrane potential of the dorsal and ventral neck motor neurons, respectively, and $\theta_{i_D}$ and $\theta_{i_V}$ represent the bias terms that shift the sensitivity of the neurotransmitter release in the dorsal and ventral neck motor neurons, respectively. $z_{i_D}$ and $z_{i_V}$ represent the strength of neurotransmitter release from the dorsal and ventral neck motor neurons, respectively.

The position ($r_x$, $r_y$) of the model worm is updated by the following equation:

$$\vec{v}(t) = \left( \frac{dr_x}{dt}, \frac{dr_y}{dt} \right) = \left( v \cos\left( \mu(t) \right), \ v \sin\left( \mu(t) \right) \right), \tag{A14}$$

where $v$ is a constant velocity of 0.022 cm/s (*Ferrée and Lockery, 1999*). The present model does not explicitly consider the kinematic mechanism that is responsible for generating the forward thrust due to the oscillatory motion of the worm. However, it is a matter of experimental observation that the locomotion of real worms never occurs without the thrust generated by undulations (*Gray and Lissmann, 1964*). In order to impose this constraint on the motion of the worm model, the fitness of the individuals that do not exhibit undulations was reduced, thus excluding such individuals from the search performed by the evolutionary algorithm (see also Evaluation of fitness).

## Numerical integration

*Equations A4, A8, A13a, and A14* were solved using the Euler integration with a time step of 0.01 s for the evolution of the network parameters and the evaluation of the chemotaxis behavior for evolved individuals. On the other hand, a time step of 0.001 s was used in the simulations that displayed the trajectories.

## Modeling of salt concentration

The salt concentrations observed during a typical salt chemotaxis assay have a Gaussian distribution (*Ward, 1973*). However, in the context of evolutionary algorithms, however, Gaussian gradients are problematic because the local steepness depends on the distance from the gradient peak. To circumvent this problem, conical gradients with randomly varying steepness were implemented in each simulation throughout the evolutionary process. Salt concentrations are proportional to the Euclidean distance from the gradient peak $\left( x_{\mathrm{peak}}, y_{\mathrm{peak}} \right)$,

$$c(t) = \alpha \sqrt{(r_x(t) - x_{\mathrm{peak}})^2 + (r_y(t) - y_{\mathrm{peak}})^2}, \qquad (\alpha < 0) \tag{A15}$$

where $\alpha$ is the slope of the gradient. Conversely, a Gaussian distribution was used to evaluate the chemotaxis behavior of most optimized individuals:

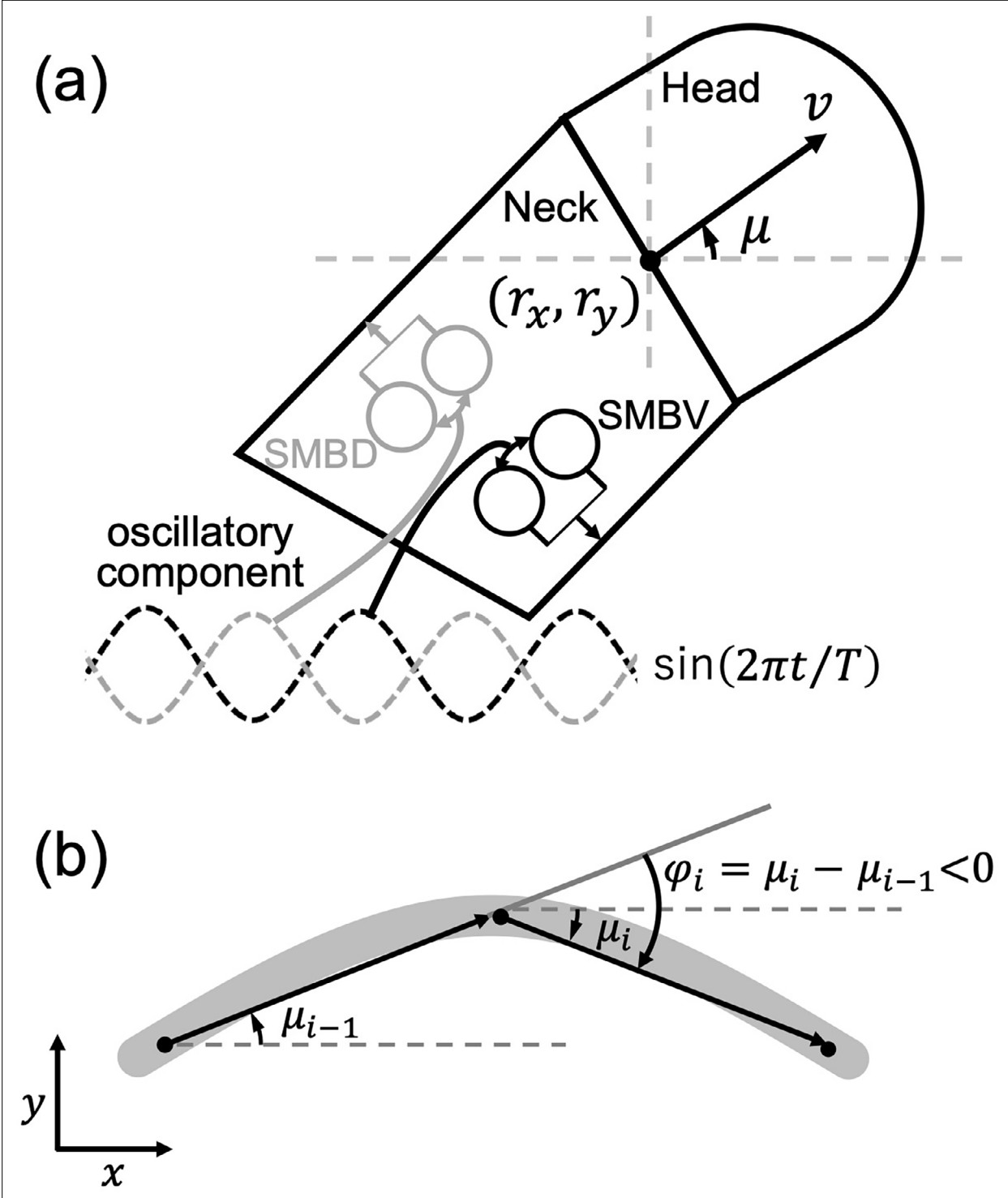

**Figure 8.** Worm locomotion model. (**a**) The body of the worm, consisting only of the idealized head and neck regions of *C. elegans*. The worm model was represented as a point $(r_x, r_y)$, located at the center of the boundary between the head and neck regions of the model. The $\mu$ represents the angle between the velocity vector *v* and the positive *x*-axis. In this context, a counterclockwise angle is considered positive. The dorsal (gray) and ventral (black) motor neuron pairs receive an out-of-phase constitutive oscillatory input from the motor systems, respectively. (**b**) Changes in the direction of locomotion. In the interval between time steps *i* −1 and *i*, the orientation of the velocity vector undergoes a change of the turning angle $\varphi_i$. The gray arc represents the path of the worm.

$$C(t) = C_0 e^{-\frac{\left(r_x(t) - x_{peak}\right)^2 + \left(r_y(t) - y_{peak}\right)^2}{2\lambda^2}}, \tag{A16}$$

where $C_0 = 1.0$ (mM) and $\lambda = 1.61$ (cm) were employed (*Ferrée and Lockery, 1999*; *Ward, 1973*).

## Evolutionary algorithm

A genetic algorithm (*Back, 1996*) was used to evolve the following parameters of the model (*Izquierdo and Beer, 2013*) in the ranges given in brackets: $w_{NMJ}$ in *Equation A13a* [1, 3]; all the biases $\theta$ [−15, 15]; the weights $w_{ji}$, $w_i^{\mathrm{ON}}$ in *Equation A7*, $w_i^{\mathrm{OFF}}$ in *Equation A7*, and $w_i^{\mathrm{self}}$ in *Equation A12a* [−15, 15]; $w_{OSC}$ in *Equation A9* [0, 15], $N$ and $M$ in *Equation A1* [0.1, 4.2]; the weight of gap junction $g_{ki}$ in *Equation A4* [0, 2.5] (*Olivares et al., 2018*). In the present study, the genetic algorithm was executed 100 times to obtain 100 optimal networks. The following is a description of a single iteration of the genetic algorithm:

1. An initial population of 60 individuals was randomly generated in which the parameters of the model were encoded in a 22-element vector of real values between [−1, 1], and these parameters were linearly mapped to their corresponding ranges.
2. The fitness of 60 individuals was quantified by using the CI that will be defined by *Equation A17*. Here, the CI for each individual was obtained as the mean CI value averaged over 50 chemotaxis assays explained later. The top 20 individuals were selected based on the fitness obtained.
3. A next generation population of 60 individuals was generated by three operations:

   i. The selected top 20 individuals were copied as the individuals comprising the next generation population.
   ii. The two-point crossover was applied, where the pairs of an odd and the next even number individual in the CI order were selected from the top 20 individuals with a probability of 0.6 as the first and second parents.
   iii. Mutations were assumed to occur in the individuals that were selected from the top 20 individuals with a probability of 0.5. The Gaussian noise with a standard deviation of 0.05 was added to the elements selected with a probability of 0.4 from the 22 elements in the vector of the mutants.
   iv. If the total number of individuals generated by the selection, crossover, and mutations was less than 60, the remaining individuals were randomly generated in the same way as in procedure 1.

4. Procedures 2 and 3 were repeated to evolve the population for 300 generations.
5. The individual with the highest-CI value after 300 generations of evolution was selected as the best performing individual.

The 100 iterations of this genetic algorithm were performed to generate the ensemble of the 100 best performing individuals. The generated top 100 individuals were ranked based on the CI value.

## Evaluation of fitness

The fitness of the individuals generated during the evolutionary algorithm was evaluated by performing the simulations with the chemotaxis assays. At the beginning of each simulation, the model worm was placed at the origin $(x, y) = (0.0, 0.0)$ with an initial orientation randomized over the range $[0.0, 2\pi]$ and motor neuron activations were randomized over the range $[0, 1]$. The gradient steepness $\alpha$ in *Equation A15* was randomized over the range $[−0.38, −0.01]$ (*Izquierdo and Beer, 2013*). Fitness was quantified as the CI, defined as the time average of the distance to the peak of the gradient at $(x_{\mathrm{peak}}, y_{\mathrm{peak}}) = (4.5, 0.0)$,

$$\mathrm{CI} = 1 - \frac{1}{T_{\mathrm{sim}}} \int_0^{T_{\mathrm{sim}}} \frac{h(t)}{h(0)} dt, \tag{A17}$$

where $h(t)$ is the Euclidean distance from the peak,

$$h(t) = \sqrt{\left(r_x(t) - x_{\mathrm{peak}}\right)^2 + \left(r_y(t) - y_{\mathrm{peak}}\right)^2}. \tag{A18}$$

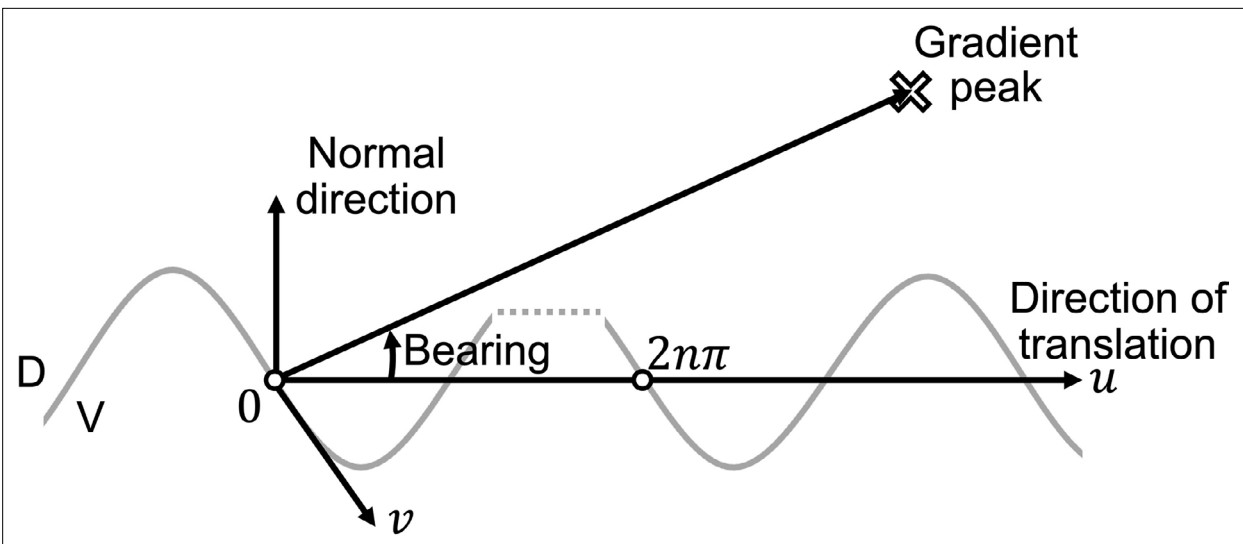

**Figure 9.** Terminology used in the analysis of the worm's locomotion. Orientation vectors used in the analysis of sinusoidal locomotion. Undulations occur in the *x*–*y* plane. The white circles represent the start and end points of *n*-cycles of locomotion, where *n* was set to three throughout the analysis of the worm's locomotion characteristics.

$h(0)$ is the initial distance of the worm from the salt peak, namely, 4.5 cm, and $T_{\text{sim}}$ is the total simulated assay time. $T_{\text{sim}}$ was set to 500 s in the evolutionary algorithm and 1000 s in the evaluation of the CI value for the individuals obtained. For simplicity, negative CI values were set to zero. The fitness of an individual generated during the evolutionary algorithm was determined as the averaged CI values over 50 trials, in which the gradient steepness $\alpha$ of the salt concentration in *Equation A15* was randomly chosen in the range of [−0.38,–0.01] and the orientation of the worm at the starting point (0.0, 0.0) was randomized (*Izquierdo and Beer, 2013*). To ensure the survival of individuals moving in an undulating manner, the CI value was reduced when sinusoidal movements were not present. Specifically, the sign of the turning angle $\varphi$ was checked at 1/4 and 3/4 cycles within each period, and if they had the same sign, the CI value was reduced by 0.008 each time as a penalty.

### Analysis of the salt preference behavior in klinotaxis

In the calculation of the quantities that were used to characterize the klinotaxis behavior, that is, curving rate, bearing, and salt concentration gradients, the coordinates of the trajectory for the first three cycles of the worm's sinusoidal locomotion were not used because the time derivative of the NaCl concentration history in *Equation A1* could not be precisely determined around the starting point, thus affecting the locomotion. The ensemble average of these quantities was determined by performing 100,000 sets of the simulation with a simulation time of $T_{\text{sim}} = 200\,\text{s}$. The Gaussian distribution of the salt concentration given by *Equation A16* was used in the analysis of the salt preference behavior in klinotaxis.

### Curving rate

Curving rate is also called turning bias (*Izquierdo and Beer, 2013*). The coordinates at three points over six cycles, namely, zero, $2n\pi$, and $4n\pi$, where $n$ is three, were used to calculate the vectors from the start point to the midpoint $\mathrm{r}(2n\pi)$ (see *Figure 9*) and from the midpoint to the end point $\mathrm{r}(4n\pi)$. The turning angle over the six cycles $\varphi(4n\pi)$ was calculated using $\mathrm{r}(2n\pi)$ and $\mathrm{r}(4n\pi)$, with counterclockwise being positive (see also *Figure 8b*, although the time interval between the vectors is different). The locomotion distance $d(4n\pi)$ was estimated from the sum of $|\mathrm{r}(2n\pi)|$ and $|\mathrm{r}(4n\pi)|$. The curving rate was determined by $\varphi(4n\pi)/d(4n\pi)$.

## Bearing

The bearing was determined as the angular difference between the vector of translational movement defined by $r(2n\pi)$ introduced above and the vector from the current position to the peak of the salt concentration gradient with counterclockwise being positive (see *Figure 9*).

## Normal and translational salt concentration gradient

The normal gradient of salt concentration was calculated from the difference in salt concentration between the position shifted by 0.001 cm toward the left perpendicular to the direction of the worm movement and the current position. The translational gradient of salt concentration was calculated from the difference in salt concentration between the position shifted by 0.001 cm toward the direction of the worm movement and the current position.

## Code availability

Full details of the methods to perform the simulations and the input data have been provided in the main text. The newly developed codes for the simulations and data analysis have been made publicly available on GitHub (copy archived at *Hironaka and Sumi, 2025*).

---

# Additional information

### Funding

| Funder | Grant reference number | Author |
|---|---|---|
| Ryobi Teien Memory Foundation | Biology Research Encouragement Award | Tomonari Sumi |
| Japan Society for the Promotion of Science | JP23K23156 | Tomonari Sumi |

The funders had no role in study design, data collection, and interpretation, or the decision to submit the work for publication.

### Author contributions

Masakatsu Hironaka, Data curation, Software, Formal analysis, Validation, Investigation, Visualization, Methodology, Writing – review and editing; Tomonari Sumi, Conceptualization, Resources, Supervision, Funding acquisition, Validation, Investigation, Visualization, Methodology, Writing – original draft, Project administration, Writing – review and editing

### Author ORCIDs

Masakatsu Hironaka ⓘ https://orcid.org/0009-0005-0194-7102
Tomonari Sumi ⓘ https://orcid.org/0000-0002-4230-5908

Reviewer #2 (Public review): https://doi.org/10.7554/eLife.104456.3.sa1
Author response https://doi.org/10.7554/eLife.104456.3.sa2

---

# Additional files

### Supplementary files

MDAR checklist

### Data availability

The experimental data shown in Figures 5d and 6a were taken from *Kunitomo et al., 2013*. The simulation results generated in this study have been made publicly available on GitHub (copy archived at *Hironaka and Sumi, 2025*).

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

# Appendix 1

**Appendix 1—table 1.** The parameters that were optimized by the genetic algorithm in the worm's chemotaxis simulation.

| Parameter | The explanation of parameter | The range of parameter |
|---|---|---|
| $N$ | The duration of the second interval over which the salt concentration is averaged in **Equation A1**. | [0.1, 4.2] (**Izquierdo and Beer, 2013**) |
| $M$ | The duration of the first interval over which the salt concentration is averaged in **Equation A1**. | [0.1, 4.2] (**Izquierdo and Beer, 2013**) |
| $\theta$ | The bias term in all the neurons in **Equations A5, A11a, A11b, A13b, and A13c**. | [−15.0, 15.0] (**Izquierdo and Beer, 2013**) |
| $w_i^{ON}$ | The strength of synaptic connection between interneuron $i$ and the ASEL sensory neurons in **Equation A7**. | [−15.0, 15.0] (**Izquierdo and Beer, 2013**) |
| $w_i^{OFF}$ | The strength of synaptic connection between interneuron $i$ and the ASER sensory neurons in **Equation A7**. | [−15.0, 15.0] (**Izquierdo and Beer, 2013**) |
| $w_{ji}$ | The strength of synaptic connection between interneurons $i$ and $j$ in **Equations A4 and A8**. | [−15.0, 15.0] (**Izquierdo and Beer, 2013**) |
| $w_i^{self}$ | The strength of self-connection of interneurons $i$ in **Equations A12a and A12b**. | [−15.0, 15.0] (**Izquierdo and Beer, 2013**) |
| $g_{ki}$ | The strength of gap junction between interneurons $k$ and $l$ in **Equation A4**. | [0.0, 2.5] (**Olivares et al., 2018**) |
| $w_{OSC}$ | The strength of the connection to the oscillatory component in **Equation A9**. | [0.0, 15.0] (**Izquierdo and Beer, 2013**) |
| $w_{NMJ}$ | The strength of the connection from motor neurons to muscles in **Equation A13a**. | [1.0, 3.0] (**Izquierdo and Beer, 2013**) |

**Appendix 1—table 2.** The parameters used in the simulations of worm's chemotaxis.

| Parameter | The explanation of parameter | The range of parameter |
|---|---|---|
| $\alpha$ | The range of randomly chosen steepness of the salt concentration gradient in **Equation A15**. | [−0.38,−0.01] (**Izquierdo and Beer, 2013**) |
| $x_{peak}$ | The $x$ coordinate of the peak position of salt concentration in **Equation A15**. | 4.5 cm (**Izquierdo and Beer, 2013**) |
| $y_{peak}$ | The $y$ coordinate of the peak position of salt concentration in **Equation A15**. | 0.0 cm (**Izquierdo and Beer, 2013**) |
| $\Delta t$ | The time step used in Euler integration of **Equations. A4, A8. A13a, and A14**. | 0.01 s was mostly used, while 0.001 s was only used to display the trajectories and determine the CI values for the most optimized networks. |
| $T$ | The duration of a one cycle of sinusoidal locomotion in **Equation A10a and A10b**. | 4.2 s (**Ferrée and Lockery, 1999**; **Izquierdo and Beer, 2013**) |
| $v$ | The velocity of locomotion in **Equation A14**. | 0.022 cm/s (**Ferrée and Lockery, 1999**; **Izquierdo and Beer, 2013**) |
| $T_{sim}$ | The total simulated assay time for the calculation of CI by **Equation A17**. | 500 s for the evolutionary algorithm (**Izquierdo and Beer, 2013**). 200 s for the analysis of klinotaxis behaviors while 1000 s for the CI calculation of obtained individuals. |
| $\tau$ | The time constant in **Equations A4 and A8**. | 0.1 s (**Izquierdo and Beer, 2013**) |
| $C_0$ | The parameter in the Gaussian distribution of salt concentration by **Equation A16** | 1.0 mM (**Ferrée and Lockery, 1999**; **Ward, 1973**) |

*Appendix 1—table 2 Continued on next page*

*Appendix 1—table 2 Continued*

| Parameter | The explanation of parameter | The range of parameter |
|---|---|---|
| λ | The parameter in the Gaussian distribution of salt concentration by *Equation A16* | 1.61 cm (*Ferrée and Lockery, 1999*; *Ward, 1973*) |

**Appendix 1—table 3.** The parameters that control the evolutionary algorithm.

| Parameter | The explanation of parameter | The range of parameter |
|---|---|---|
| average | The number of simulations that were performed to average the CI values of an individual. | 50 (*Izquierdo and Beer, 2013*) |
| gen_size | The length of gene that was equal to the number of parameters that were searched. | 22 This work |
| ga_count | The number of iterations for which the evolutionary algorithm was run. | 100 (*Izquierdo and Beer, 2013*) |
| n_gen | The number of generations for which the populations evolved. | 300 (*Izquierdo and Beer, 2013*) |
| pop_size | The number of individuals that were included in the populations | 60 (*Izquierdo and Beer, 2013*) |
| sel_top | The number of top individuals that were selected. | 20 This work |
| mat_pb | The probability by which a pair of parents was selected for a two-point crossover. | 0.6 This work |
| mut_pb | The probability by which a mutant was selected. | 0.5 This work |
| mut_in_pb | The probability with which a mutation occurs to the 22 elements in the vector of a selected mutant. | 0.4 This work |

**Appendix 1—table 4.** The parameters of the most optimized network model with the constraints, as obtained from the evolutionary algorithm with the assumption that the AIY–AIZ connections are inhibitory.

| Parameter | The optimized parameter |
|---|---|
| $N$ (s) | $N = 0.4907$ |
| $M$ (s) | $M = 0.7618$ |
| $\theta$ | $\theta_{\mathrm{AIYL}} = 0.8839$ $\theta_{\mathrm{AIYR}} = -7.3416$ $\theta_{\mathrm{AIZL}} = 2.3906$ $\theta_{\mathrm{AIZR}} = 5.3649$ $\theta_{\mathrm{SMBDL}} = \theta_{\mathrm{SMBVL}} = -8.4964$ $\theta_{\mathrm{SMBDR}} = \theta_{\mathrm{SMBVR}} = -11.7800$ |
| $w_i^{\mathrm{ON}}$ | $w_{\mathrm{AIYL}}^{\mathrm{ON}} = 9.8280$ $w_{\mathrm{AIYR}}^{\mathrm{ON}} = -9.7935$ |

*Appendix 1—table 4 Continued on next page*

*Appendix 1—table 4 Continued*

| Parameter | The optimized parameter |
|---|---|
| $w_i^{\text{OFF}}$ | $w_{\text{AIYL}}^{\text{OFF}} = 8.2233$<br>$w_{\text{AIYR}}^{\text{OFF}} = -14.3481$ |
| | $w_{\text{AIYL,AIZL}} = -15.0000$<br>$w_{\text{AIYR,AIZR}} = -11.0792$<br><br>$w_{\text{AIZL,SMBDL}} = w_{\text{AIZL,SMBVL}} = 0.3112$<br><br>$w_{\text{AIZR,SMBDR}} = w_{\text{AIZR,SMBVR}} = 10.7255$ |
| $w_i^{\text{self}}$ | $w_{\text{SMBDL}}^{\text{self}} = w_{\text{SMBVL}}^{\text{self}} = -13.8653$<br>$w_{\text{SMBDR}}^{\text{self}} = w_{\text{SMBVR}}^{\text{self}} = 2.0301$ |
| $g_{ki}$ | $g_{\text{AIYL,AIYR}} = g_{\text{AIYR,AIYL}} = 2.4368$<br>$g_{\text{AIZL,AIZR}} = g_{\text{AIZR,AIZL}} = 2.2160$ |
| $w_{\text{OSC}}$ | $w_{\text{OSE}} = 2.9655$ |
| $w_{\text{NMJ}}$ | $w_{\text{NMJ}} = 2.7969$ |

