## [Editor Report · eLife Assessment]

With a computational analysis of a neuroanatomical network model in *C. elegans*, this **valuable** work investigates the synaptic mechanism for memory-dependent klinotaxis, i.e., salt concentration chemotaxis. By incorporating experimental data altering the ASER neuron's basal glutamate release into their model, the authors demonstrate the possibility of a transition between excitatory and inhibitory signaling at the ASER-AIY synapse, depending on environmental and cultivated salt concentrations. These **solid** findings offer a proposal for how synaptic plasticity plays a role in sensorimotor navigation, and will be of interest to worm biologists and theoretical neuroscientists.

---

## [Referee Report · Reviewer #2 (Public review)]

Summary:

This study explores how a simple sensorimotor circuit in the nematode *C. elegans* enables it to navigate salt gradients based on past experiences. Using computational simulations and previously described neural connections, the study demonstrates how a single neuron, ASER, can change its signaling behavior in response to different salt conditions, with which the worm is able to "remember" prior environments and adjust its navigation toward "preferred" salinity accordingly.

Strengths:

The key novelty and strength of this paper is the explicit demonstration of computational neurobehavioral modeling and evolutionary algorithms to elucidate the synaptic plasticity in a minimal neural circuit that is sufficient to replicate memory-based chemotaxis. In particular, with changes in ASER's glutamate release and sensitivity of downstream neurons, the ASER neuron adjusts its output to be either excitatory or inhibitory depending on ambient salt concentration, enabling the worm to navigate toward or away from salt gradients based on prior exposure to salt concentration.

Weaknesses:

While the model successfully replicates some behaviors observed in previous experiments, some key assumptions of the work still need to be verified by biological validation of further experiments.

Comments on revisions:

Thank you for the authors' response. The revision and their response have substantially addressed my concerns.

---

## [Author Response]

The following is the authors’ response to the original reviews

**eLife Assessment**
The authors utilize a valuable computational approach to exploring the mechanisms of memorydependent klinotaxis, with a hypothesis that is both plausible and testable. Although they provide a solid hypothesis of circuit function based on an established model, the model's lack of integration of newer experimental findings, its reliance on predefined synaptic states, and oversimplified sensory dynamics, make the investigation incomplete for both memory and internal-state modulation of taxis.

We would like to express our gratitude to the editor for the assessment of our work. However, we respectfully disagree with the assessment that our investigation is incomplete, if the negative assessment is primarily due to the impact of AIY interneuron ablation on the chemotaxis index (CI) which was reported in Reference [1]. It is crucial to acknowledge that the CI determined through experimental means incorporates contributions from both klinokinesis and klinotaxis [1]. It is plausible that the impact of AIY ablation was not adequately reflected in the CI value. Consequently, the experimental observation does not necessarily diminish the role of AIY in klinotaxis. Anatomical evidence provided by the database (http://ims.dse.ibaraki.ac.jp/ccep-tool/) substantiates that ASE sensory neurons and AIZ interneurons, which have been demonstrated to play a crucial role in klinotaxis [Matsumoto et al., PNAS 121 (5) e2310735121], have the much higher number of synaptic connections with AIY interneurons. These findings provide substantial evidence supporting the validity of the presented minimal neural network responsible for salt klinotaxis.

**Public Reviews:**

**Reviewer #1 (Public review):**
Summary:This research focuses on *C. elegans* klinotaxis, a chemotactic behavior characterized by gradual turning, aiming to uncover the neural circuit mechanism responsible for the context-dependent reversal of salt concentration preference. The phenomenon observed is that the preferred salt concentration depends on the difference between the pre-assay cultivation conditions and the current environmental salt levels.

We would like to express our gratitude for the time and consideration you have dedicated to reviewing our manuscript.

The authors propose that a synaptic-reversal plasticity mechanism at the primary sensory neuron, ASER, is critical for this memory- and context-dependent switching of preference. They build on prior findings regarding synaptic reversal between ASER and AIB, as well as the receptor composition of AIY neurons, to hypothesize that similar "plasticity" between ASER and AIY underpins salt preference behavior in klinotaxis. This plasticity differs conceptually from the classical one as it does not rely on any structural changes but rather synaptic transmission is modulated by the basal level of glutamate, and can switch from inhibitory to excitatory.To test this hypothesis, the study employs a previously established neuroanatomically grounded model [4] and demonstrates that reversing the ASER-AIY synapse sign in the model agent reproduces the observed reversal in salt preference. The model is parameterized using a computational search technique (evolutionary algorithm) to optimize unknown electrophysiological parameters for chemotaxis performance. Experimental validity is ensured by incorporating constraints derived from published findings, confirming the plausibility of the proposed mechanism.Finally. the circuit mechanism allowing *C. elegans* to switch behaviour to an exploration run when starved is also investigated. This extension highlights how internal states, such as hunger, can dynamically reshape sensory-motor programs to drive context-appropriate behaviors.

We would like to thank the reviewer for the appropriate summary of our work.

Strengths and weaknesses:The authors' approach of integrating prior knowledge of receptor composition and synaptic reversal with the repurposing of a published neuroanatomical model [4] is a significant strength. This methodology not only ensures biological plausibility but also leverages a solid, reproducible modeling foundation to explore and test novel hypotheses effectively.The evidence produced that the original model has been successfully reproduced is convincing.The writing of the manuscript needs revision as it makes comprehension difficult.

We would like to thank the reviewer for recognizing the usefulness of our approach. In the revised version, we improved the explanation according to your suggestions.

One major weakness is that the model does not incorporate key findings that have emerged since the original model's publication in 2013, limiting the support for the proposed mechanism. In particular, ablation studies indicate that AIY is not critical for chemotaxis, and other interneurons may play partially overlapping roles in positive versus negative chemotaxis. These findings challenge the centrality of AIY and suggest the model oversimplifies the circuit involved in klinotaxis.

We would like to express our gratitude for the constructive feedback we have received. We concur with some of your assertions. In fact, our model is the minimal network for salt klinotaxis, which includes solely the interneurons that are connected to each other via the highest number of synaptic connections. It is important to note that our model does not consider redundant interneurons that exhibit overlapping roles. Consequently, the model is not applicable to the study of the impact of interneuron ablation. In the reference [1], the influence of interneuron ablations on the chemotaxis index (CI) has been investigated. The experimentally determined CI value incorporates the contributions from both klinokinesis and klinotaxis. Consequently, it is plausible that the impact of AIY ablation was not significantly reflected in the CI value. The experimental observation does not necessarily diminish the role of AIY in klinotaxis.

Reference [1] also shows that ASER neurons exhibit complex, memory- and context-dependent responses, which are not accounted for in the model and may have a significant impact on chemotactic model behaviour.

As the reviewer has noted, our model does not incorporate the context-dependent response of the ASER. Instead, the impact of the salt concentration-dependent glutamate release from the ASER [S. Hiroki et al. Nat Commun 13, 2928 (2022)] as the result of the ASER responses was in detail examined in the present study.

The hypothesis of synaptic reversal between ASER and AIY is not explicitly modeled in terms of receptor-specific dynamics or glutamate basal levels. Instead, the ASER-to-AIY connection is predefined as inhibitory or excitatory in separate models. This approach limits the model's ability to test the full range of mechanisms hypothesized to drive behavioral switching.

We would like to express our gratitude to the reviewer for their constructive feedback. As you correctly noted, the hypothesized synaptic reversal between ASER and AIY is not explicitly modeled in terms of the sensitivity of the receptors in the AIY and the glutamate basal levels by the ASER. On the other hand, in the present study, under considering a substantial difference in the sensitivity of the two glutamate receptors on the AIY, we sought to endeavored to elucidate the impact of salt-concentration-dependent glutamate basal levels on klinotaxis. To this end, we conducted a comprehensive examination of the full range gradual change in the ASER-to-AIY connection from inhibitory to excitatory, as illustrated in Figures S4 and S5.

While the main results - such as response dependence on step inputs at different phases of the oscillator - are consistent with those observed in chemotaxis models with explicit neural dynamics (e.g., Reference [2]), the lack of richer neural dynamics could overlook critical effects. For example, the authors highlight the influence of gap junctions on turning sensitivity but do not sufficiently analyze the underlying mechanisms driving these effects. The role of gap junctions in the model may be oversimplified because, as in the original model [4], the oscillator dynamics are not intrinsically generated by an oscillator circuit but are instead externally imposed via $z_¥text{osc}$. This simplification should be carefully considered when interpreting the contributions of specific connections to network dynamics. Lastly, the complex and contextdependent responses of ASER [1] might interact with circuit dynamics in ways that are not captured by the current simplified implementation. These simplifications could limit the model's ability to account for the interplay between sensory encoding and motor responses in *C. elegans* chemotaxis.

We might not understand the substance of your assertions. However, we understand that the oscillator dynamics were not intrinsically generated by the oscillator neural circuit that is explicitly incorporated into our modeling. On the other hand, the present study focuses on how the sensory input and resulting interneuron dynamics regulate the oscillatory behavior of SMB motor neurons to generate klinotaxis. The neuron dynamics via gap junctions results from the equilibration of the membrane potential *yi* of two neurons connected by gap junctions rather than the *zi*. We added this explanation in the revised manuscript as follows.

“The hyperpolarization signals in the AIZL are transmitted to the AIZR via the gap junction (Figs. S1d and S1f and Fig. 3d). This is because the neuron dynamics via gap junctions results from the equilibration of the membrane potential *yi* of two neurons connected by gap junctions rather than the *zi*.”

In the limitation, we added the following sentence:

“In the present study, the oscillator components of the SMB are not intrinsically generated by an oscillator circuit but are instead externally imposed via *𝑧i*^OSC^. Furthermore, the complex and context-dependent responses of ASER {Luo:2014et} were not taken into consideration. It should be acknowledged as a limitation of this study that these omitted factors may interact with circuit dynamics in ways that are not captured by the current simplified implementation.”

Appraisal:The authors show that their model can reproduce memory-dependent reversal of preference in klinotaxis, demonstrating that the ASER-to-AIY synapse plays a key role in switching chemotactic preferences. By switching the ASER-AIY connection from excitatory to inhibitory they indeed show that salt preference reverses. They also show that the curving/turn rate underlying the preference change is gradual and depends on the weight between ASER-AIY. They further support their claim by showing that curving rates also depend on cultivated (set-point).

We would like to thank the reviewer for assessing our work.

Thus within the constraints of the hypothesis and the framework, the model operates as expected and aligns with some experimental findings. However, significant omissions of key experimental evidence raise questions on whether the proposed neural mechanisms are sufficient for reversal in salt-preference chemotaxis.

We agree with your opinion. The present hypothesis should be verified by experiments.

Previous work [1] has shown that individually ablating the AIZ or AIY interneurons has essentially no effect on the Chemotactic Index (CI) toward the set point ([1] Figure 6). Furthermore, in [1] the authors report that different postsynaptic neurons are required for movement above or below the set point. The manuscript should address how this evidence fits with their model by attempting similar ablations. It is possible that the CI is rescued by klinokinesis but this needs to be tested on an extension of this model to provide a more compelling argument.

We would like to express our gratitude for the constructive feedback we have received. In the reference [1], the influence of interneuron ablations on the chemotaxis index (CI) has been investigated. It is important to acknowledge that the experimentally determined CI value encompasses the contributions of both klinokinesis and klinotaxis. It is plausible that the impact of AIY ablation was not reflected in the CI value. Consequently, these experimental observations do not necessarily diminish the role of AIY in klinotaxis. The neural circuit model employed in the present study constitutes a minimal network for salt klinotaxis, encompassing solely interneurons that are connected to each other via the highest number of synaptic connections. Anatomical evidence provided by the database (http://ims.dse.ibaraki.ac.jp/cceptool/) substantiates that ASE sensory neurons and AIZ interneurons, which have been demonstrated to play a crucial role in klinotaxis [Matsumoto et al., PNAS 121 (5) e2310735121], have the much higher number of synaptic connections with AIY interneurons. Our model does not take into account redundant interneurons with overlapping roles, thus rendering it not applicable to the study of the effects of interneuron ablation.

The investigation of dispersal behaviour in starved individuals is rather limited to testing by imposing inhibition of the SMB neurons. Although a circuit is proposed for how hunger states modulate taxis in the absence of food, this circuit hypothesis is not explicitly modelled to test the theory or provide novel insights.

As the reviewer noted, the experimentally identified neural circuit that inhibits the SMB motor neurons in starved individuals is not incorporated in our model. Instead of incorporating this circuit explicitly, we examined whether our minimal network model could reproduce dispersal behavior under starvation conditions solely due to the experimentally demonstrated inhibitory effect of SMB motor neurons.

Impact:This research underscores the value of an embodied approach to understanding chemotaxis, addressing an important memory mechanism that enables adaptive behavior in the sensorimotor circuits supporting *C. elegans* chemotaxis. The principle of operation - the dependence of motor responses to sensory inputs on the phase of oscillation - appears to be a convergent solution to taxis. Similar mechanisms have been proposed in Drosophila larvae chemotaxis [2], zebrafish phototaxis [3], and other systems. Consequently, the proposed mechanism has broader implications for understanding how adaptive behaviors are embedded within sensorimotor systems and how experience shapes these circuits across species.

We would like to express our gratitude for useful suggestion. We added this argument in Discussion of the revised manuscript as follows.

“The principle of operation, in which the dependence of motor responses to sensory inputs on the phase of motor oscillation, appears to be a convergent solution for taxis and navigation across species. In fact, analogous mechanisms have been postulated in the context of chemotaxis in *Drosophila* larvae chemotaxis {Wystrach:2016bt} and phototaxis in zebrafish {Wolf:2017ei}. Consequently, the synaptic reversal mechanism highlighted in this study offers the framework for understanding how the behaviors that are adaptive to the environment are embedded within sensorimotor systems and how experience shapes these neural circuits across species.”

Although the reported reversal of synaptic connection from excitatory to inhibitory is an exciting phenomenon of broad interest, it is not entirely new, as the authors acknowledge similar reversals have been reported in ASER-to-AIB signaling for klinokinesis (Hiroki et al., 2022). The proposed reversal of the ASER-to-AIY synaptic connection from inhibitory to excitatory is a novel contribution in the specific context of klinotaxis. While the ASER's role in gradient sensing and memory encoding has been previously identified, the current paper mechanistically models these processes, introducing a hypothesis for synaptic plasticity as the basis for bidirectional salt preference in klinotaxis.The research also highlights how internal states, such as hunger, can dynamically reshape sensory-motor programs to drive context-appropriate behaviors.The methodology of parameter search on a neural model of a connectome used here yielded the valuable insight that connectome information alone does not provide enough constraints to reproduce the neural circuits for behaviour. It demonstrates that additional neurophysiological constraints are required.

We would like to acknowledge the appropriate recognition of our work.

Additional ContextOscillators with stimulus-driven perturbations appear to be a convergent solution for taxis and navigation across species. Similar mechanisms have been studied in zebrafish phototaxis [3], Drosophila larvae chemotaxis [2], and have even been proposed to underlie search runs in ants. The modulation of taxis by context and memory is a ubiquitous requirement, with parallels across species. For example, Drosophila larvae modulate taxis based on current food availability and predicted rewards associated with odors, though the underlying mechanism remains elusive. The synaptic reversal mechanism highlighted in this study offers a compelling framework for understanding how taxis circuits integrate context-related memory retrieval more broadly.

We would like to express our gratitude for the insightful commentary. In the revised manuscript, we incorporated the argument that the similar oscillator mechanism with stimulus-driven perturbations has been observed for zebrafish phototaxis [3] and Drosophila larvae chemotaxis [2] into Discussion.

As a side note, an interesting difference emerges when comparing *C. elegans* and Drosophila larvae chemotaxis. In Drosophila larvae, oscillatory mechanisms are hypothesized to underlie all chemotactic reorientations, ranging from large turns to smaller directional biases (weathervaning). By contrast, in *C. elegans*, weathervaning and pirouettes are treated as distinct strategies, often attributed to separate neural mechanisms. This raises the possibility that their motor execution could share a common oscillator-based framework. Re-examining their overlap might reveal deeper insights into the neural principles underlying these maneuvers.

We would like to acknowledge your thoughtfully articulated comment. As the reviewer pointed out, the anatomical database (http://ims.dse.ibaraki.ac.jp/ccep-tool/) shows that that the neural circuits underlying weathervaning and pirouettes in *C. elegans* are predominantly distinct but exhibit partial overlap. When we restrict our search to the neurons that are connected to each other with the highest number of synaptic connections, we identify the projections from the neural circuit of weathervaning to the circuit of pirouettes; however we observed no reversal projections. This finding suggests that the neural circuit of weathervaning, namely, our minimal neural network, is not likely to be affected by that of pirouettes, which consists of AIB interneurons and interneurons and motor neurons the downstream.

(1) Luo, L., Wen, Q., Ren, J., Hendricks, M., Gershow, M., Qin, Y., Greenwood, J., Soucy, E.R., Klein, M., Smith-Parker, H.K., & Calvo, A.C. (2014). Dynamic encoding of perception, memory, and movement in a *C. elegans* chemotaxis circuit. Neuron, 82(5), 1115-1128.

(2) Antoine Wystrach, Konstantinos Lagogiannis, Barbara Webb (2016) Continuous lateral oscillations as a core mechanism for taxis in Drosophila larvae eLife 5:e15504.

(3) Wolf, S., Dubreuil, A.M., Bertoni, T. et al. Sensorimotor computation underlying phototaxis in zebrafish. Nat Commun 8, 651 (2017).

(4) Izquierdo, E.J. and Beer, R.D., 2013. Connecting a connectome to behavior: an ensemble of neuroanatomical models of *C. elegans* klinotaxis. PLoS computational biology, 9(2), p.e1002890.

**Reviewer #2 (Public review):**
Summary:This study explores how a simple sensorimotor circuit in the nematode *C. elegans* enables it to navigate salt gradients based on past experiences. Using computational simulations and previously described neural connections, the study demonstrates how a single neuron, ASER, can change its signaling behavior in response to different salt conditions, with which the worm is able to "remember" prior environments and adjust its navigation toward "preferred" salinity accordingly.

We would like to express our gratitude for the time and consideration the reviewer has dedicated to reviewing our manuscript.

Strengths:The key novelty and strength of this paper is the explicit demonstration of computational neurobehavioral modeling and evolutionary algorithms to elucidate the synaptic plasticity in a minimal neural circuit that is sufficient to replicate memory-based chemotaxis. In particular, with changes in ASER's glutamate release and sensitivity of downstream neurons, the ASER neuron adjusts its output to be either excitatory or inhibitory depending on ambient salt concentration, enabling the worm to navigate toward or away from salt gradients based on prior exposure to salt concentration.

We would like to thank the reviewer for appreciating our research.

Weaknesses:While the model successfully replicates some behaviors observed in previous experiments, many key assumptions lack direct biological validation. As to the model output readouts, the model considers only endpoint behaviors (chemotaxis index) rather than the full dynamics of navigation, which limits its predictive power. Moreover, some results presented in the paper lack interpretation, and many descriptions in the main text are overly technical and require clearer definitions.

We would like to thank the reviewer for the constructive feedback. As the reviewer noted, the fundamental assumptions posited in the study have yet to be substantiated by biological validation, and consequently, these assumptions must be directly assessed by biological experimentation. The model performance for salt klinotaxis has been evaluated by multiple factors, including not only a chemotaxis index but also the curving rate vs. bearing (Fig. 4a, the bearing is defined in Fig. A3) and the curving rate vs. normal gradient (Fig. 4c). These two parameters work to characterize the trajectory during salt klinotaxis. In the revised version, we meticulously revised the manuscript according to the reviewer’s suggestions. We would like to express our sincere gratitude for your insightful review of our work.

**Recommendations for the authors:**

**Reviewer #1 (Recommendations for the authors):**
An interesting and engaging methodology combining theoretical and computational approaches. Overall I found the manuscript up to discussion a difficult read, and I would suggest revising it. I would also recommend introducing the general operating principle of the oscillator with sensory perturbations before jumping into the implementation details of signal propagation specific to C.

elegans.

In order to elucidate the relation between the general operating principle of the oscillator with sensory perturbations and the results shown by the two graphs from the bottom in Fig. 3d, the following statement was added on page 12.

“It is remarkable that this regulatory mechanism derived via the optimization of the CI has been observed in the context of chemotaxis in *Drosophila* larvae chemotaxis {Wystrach:2016bt} and phototaxis in zebrafish {Wolf:2017ei}. The principle of operation, in which the dependence of motor responses to sensory inputs on the phase of motor oscillation, therefore, may serve as a convergent solution for taxis and navigation across species.”

The abstract could benefit from a clarification of terms to benefit a broader audience: The term "salt klinotaxis" is used without prior introduction or definition. It would be beneficial to briefly explain this term, as it may not be familiar to all readers.

Due to the limitation of the word number in the abstract, the explanation of salt klinotaxis could not be included.

Although ASER is introduced as a right-side head sensory neuron, AIY neurons are not similarly introduced. It may also benefit to introduce here that ASER integrates memory with current salt gradients, tuning its output to produce context-appropriate behaviour.

Due to the limitation of the word number in the abstract, we could add no more the explanations.

"it can be anticipated that the ASER-AIY synaptic transmission will undergo a reversal due to alterations in the basal glutamate Release": Where is this expectation drawn from? Is it derived from biophysical or is it a functional expectation to explain the network's output constraints?

As delineated before this sentence, it is derived from a comprehensive consideration of the sensitivity of excitatory/inhibitory glutamate receptors expressed on the postsynaptic AIY interneurons, in conjunction with varying the basal level of glutamate transmission from ASER.

The statement that the model "revealed the modular neural circuit function downstream of ASE" could be more explicit. What specific insights about the downstream circuit were uncovered?Highlighting one or two key findings would strengthen the impact.

Due to the limitation of the word number in the abstract, no more details could be added here, while the sentence was revised as “revealed that the circuit downstream of ASE functions as a module that is responsible for salt klinotaxis.” This is because the salt-concentration dependent behaviors in klinitaxis can be reproduced through the modulation of the ASRE-AIY synaptic connections alone, despite the absence of alterations in the neural circuit downstream of AIY.

I believe the authors should cite Luo et al. 2014, which also studies how chemotactic behaviours arise from neural circuit dynamics, including the dynamic encoding of salt concentration by ASER, and the crucial downstream interaction with AIY for chemotactic actions.

We would like to express our gratitude for useful suggestion. We cited Luo et al. 2014 in the discussion on the limitation of our work.

The introduction could also be improved for clarity. Specifically in the last paragraph authors should clarify how the observed synchrony of ASER excitation to the AIZ (Matsumoto et al., 2024), validates the resulting network.

We would like to express our gratitude for useful suggestion. We added the following explanation in the last paragraph of the introduction.

“Specifically, the synchrony of the excitation of the ASER and AIZ {Matsumoto:2024ig} taken together with the experimentally identified inhibitory synaptic transmission between the AIY and AIZ revealed that the ASER-AIY synaptic connections should be inhibitory, which was consistent with the network obtained from the most evolved model.”

In addition, we added the following explanation after “It was then hypothesized that the ASER-AIY inhibitory synaptic connections are altered to become excitatory due to a decrease in the baseline release of glutamate from the ASER when individuals are cultured under *C*_cult_ < *C*_test_.”

This is due to the substantial difference in the sensitivity of excitatory/inhibitory glutamate receptors expressed on the postsynaptic AIY interneurons.

I would also strongly recommend replacing the term "evolved model", with "Optimized Model" or "Best-Performing Model" to clarify this is a computational optimization process with limitations - optimization through GAs does not guarantee finding global optima.

We revised "evolved model" as "optimized model" in the main and SI text.

The text overall would benefit from editing for clarity and expression.

According to the revisions mentioned above, we revised “best optimized model” as “most optimized model” in the main and SI text.

The font size on the plot axis in Figures 3 c&d should be increased for readability on the printed page. Label the left/right panel to indicate unconstrained / constrained evolution.

As you noted, the font size of the subscript on the vertical axis in Figs 3c and 3d was too small. We have revised the font size of the subscript in Figs. 3c and 3d and also in Fig. 5e. At your suggestion, “unconstrained” and “constrained” have been added as labels to the left and right panels in Fig. 3.

There is no input/transmission to AIYR to step input in either model shown in Figure 3?

As shown in Fig. S1e and S1f, there are the transmissions to the AIYR from the ASEL and ASER.

Supplementary Figure 1 attempts to explain the interactions. There are inconsistent symbols used for inhibition and excitation between network schema (colours) and the z response plots (arrows vs circles), combined with different meanings for red/blue making it very confusing.

We could not address the inconsistency in the color of arrows and lines with an ending between Figs. S1c and S1d and Figs. S1a and S1b. On the other hand, Figs. S1e and S1f were revised so that the consistent symbols were used for inhibition, excitation, and electrical gap connections in Figs. S1c-S1f. The same revisions were made for Fig. S7c-S7f.

Model parameters are given to 15 decimal precision, which seems excessive. Is model performance sensitive to that order? We would expect robustness around those values. The authors should identify relevant orders and truncate parameters accordingly.

We examined the influence of the parameter truncation on the trajectory and decided that the parameters with four decimal places were appropriate. According to this, we revised Table A4.

Figure 3 caption typo "step changes I the salt concentration".

The typo was revised in Fig. 3 caption.

**Reviewer #2 (Recommendations for the authors):**
(1) Overall, the language of the paper is not properly organized, making the paper's logic and purpose hard to follow. In the Results Section, many observations or findings lack explicit interpretation. To address this issue, the authors should consider (1) adopting the contextcontent-conclusion scheme, (2) optimizing the logic flow by clearly identifying the context and goals prior to discussing their results and findings, (3) more explicitly interpreting their results, especially in a biological context.

We would like to express our gratitude for helpful suggestion. According to your suggestion listed below, we revised the main and SI texts.

(2) In Figure 2, trajectories from the model with AIY-AIZ constraints show a faster convergence than those from the constraint-free model. However, in the corresponding texts in the Results section, the authors claimed no significant difference. It seems that the authors made this argument only based on CI (Chemotaxis Index). Therefore, in order to address such inconsistency, the authors need more explanation on why only relying on CI, which is an endpoint metric, instead of the whole navigation.

I would like to thank you for the helpful comment. In the present study, not only the CI but also the curving rate shown in Fig. 4 were applied to characterize the behavior in klinotaxis.

According to your comments, we revised the related description in the main text as follows:

“The difference between these CI values is slight, while the model optimized with the constraints exhibits a marginally accelerated attainment of the salt concentration peak, as shown by the trajectories. The slightly higher chemotaxis performance observed in the constrained model is not essentially attributed to the introduction of the AIY-AIZ synaptic constraints but rather depends on the specific individuals selected from the optimized individuals obtained from the evolutionary algorithm. In fact, even when the AIY-AIZ constraints are taken into consideration, the model retains a significant degree of freedom to reproduce salt klinotaxis due to the presence of a substantial parameter space. Consequently, the impact of the AIY-AIZ constraints on the optimization of the CI is expected to be negligible.”

(3) In Figures 3a and b, some inter-neuron connections are relatively weak (e.g., AIYR to AIZR in Figure 3a) - thus it is unclear whether the polarity of such synapses would significantly influence the behavioral outcome or not. The authors could consider plotting the change of the connection strengths between neurons over the course of model optimization to get a sense of confidence in each inter-neuron connection.

In the evolutional algorithm, the parameters of individuals are subject to discontinuous variation due to the influence of selection, crossover, and mutations. Consequently, it is not straightforward to extract information regarding parameter optimization from parameter changes due to the non-systematic nature of parameter variation..

(4) In Figure 3, the order of individual figure panels is incorrect: in the main text, Figure 3 a and b were mentioned after c and d. Also, the caption of Figure 3c "negative step changes I the" should be "in".

The main text underwent revision, with the description of Figures 3a and 3b being presented prior to that of Figures 3c and 3d. The typo was revised.

(5) In Figure 4, the order of individual figure panels is messed up: in the main text, Figure 4 a was mentioned after b.

The main text underwent revision, with the description of Figure 4a being presented prior to that of Figure 4b.

(6) Also in Figure 4, the authors need to provide a definition/explanation of "Bearing" and "Translational Gradient". In Figure 4d, the definition of positive and negative components is not clear.

Normal and Translational Salt Concentration Gradient in METHOD was referenced for the definition and explanation of the bearing and the translational gradient. We added the following explanation on the positive and negative components.

“The positive and negative components of the curving rate are respectively sampled from the trajectory during leftward turns (as illustrated in Fig. 4b) and rightward turns, respectively.”

(7) Figure 5: the authors need to explain why c has an error bar and how they were calculated, as this result is from a computational model. Figure 5d is experimental results - the authors need to add error bars to the data points and provide a sample size.

As explained in Analysis of the Salt Preference Behavior in Klinotaxis in METHOD, the ensemble average of these quantities was determined by performing 100,000 sets of the simulation with randomized initial orientation for a simulation time of T_sim=200 sec. The error bars for the experimental data were added in Figs. 5c, 6a, and S9a.

(8) On Page 14, the authors said, "To this end, this end, we used the best evolved network with the constraints, in which we varied the synaptic connections between ASER and AIY from inhibitory to excitatory." How did the model change the ASER-AIY signaling specifically? The authors should provide more explanation or at least refer to the Methods Section.

The caption of Fig. S4 was referred as the explanation on the detailed method.

(9) Page 15: "a subset a subset exhibited a slight curve...". This observation from the model simulation is contradictory to experiments. However, their explanation of that is hard to understand.

I would like to thank you for the helpful comment. To improve this, we added the following explanation:

“In the case of step increases in 𝑧OFF as illustrated in the second right panel from the bottom in Fig.3d, the turning angle φ is increased from its ideal oscillatory component to a value close to zero, causing the model worm to deviate from the ideal sinusoidal trajectory and gradually turn toward lower salt concentrations. On the other hand, in the case of step increases in 𝑧ON as illustrated in the second left panel from the bottom in Fig.3d, the turning angle φ is again increased from its ideal oscillatory component to a value close to zero, causing the model worm to deviate from the ideal sinusoidal trajectory and gradually turn toward higher salt concentrations. The behaviors that are consistent with these analyses are observed in the trajectory illustrated in Fig. S8b.”

(10) Last result session: inhibited SMB in starved worms is due to a mechanism unrelated to their neural network model upstream to SMB. Therefore, their results recapitulating the worms' dispersal behaviors cannot strengthen the validity of their model.

We agree with your opinion. We think that the findings from the study of starved worms do not provide evidence to validate the neural network model upstream of SMB.

(11) Discussion: "in contrast, the remaining neurons...". This argument lacks evidence or references.

This argument is based on the results obtained from the present study. This sentence was revised as follows:

“This regulatory process enables the reproduction of salt concentration memory-dependent reversal of preference behavior in klinotaxis, despite the remaining neurons further downstream of the ASER not undergoing alterations and simply functioning as a modular circuit to transmit the received signals to the motor systems. Consequently, the sensorimotor circuit allows a simple and efficient bidirectional regulation of salt preference behavior in klinotaxis.”

(12) To increase the predictive power of their model, can the authors perform simulations on mutant worms, like those with altered glutamate basal level expression in ASER?

We would like to express our gratitude for useful suggestion. The simulations, in which the weight of the ASER-AIY synaptic connection is increased from negative (inhibitory connection) to positive (excitatory connection), as illustrated in Figure S4, provide valuable insights into the relationship between varying glutamate basal levels from ASER and behavior in klinotaxis, such as the chemotaxis index.